# Electrocatalytic synthesis of heterocycles from biomass-derived furfuryl alcohols

Xuan Liu [1], Bo Li[2], Guanqun Han[1], Xingwu Liu[3], Zhi Cao[3,4 ✉], De-en Jiang [2 ✉] & Yujie Sun [1 ✉]

It is very attractive yet underexplored to synthesize heterocyclic moieties pertaining to biologically active molecules from biomass-based starting compounds. Herein, we report an electrocatalytic Achmatowicz reaction for the synthesis of hydropyranones from furfuryl alcohols, which can be readily produced from biomass-derived and industrially available furfural. Taking advantage of photo-induced polymerization of a bipyridyl ligand, we demonstrate the facile preparation of a heterogenized nickel electrocatalyst, which effectively drives the Achmatowicz reaction electrochemically. A suite of characterization techniques and density functional theory computations were performed to aid the understanding of the reaction mechanism. It is rationalized that the unsaturated coordination sphere of nickel sites in our electrocatalyst plays an important role at low applied potential, not only allowing the intimate interaction between the nickel center and furfuryl alcohol but also enabling the transfer of hydroxide from nickel to the bound furfuryl alcohol.

[1] Department of Chemistry, University of Cincinnati, Cincinnati, OH, USA. [2] Department of Chemistry, University of California, Riverside, CA, USA. [3] Syncat@Beijing, Synfuels CHINA Co., Ltd, Beijing, China. [4] State Key Laboratory of Coal Conversion, Institute of Coal Chemistry, Chinese Academy of Sciences, Taiyuan, China. ✉email: caozhi@sxicc.ac.cn; djiang@ucr.edu; yujie.sun@uc.edu

In light of the diminishing fossil reserves and the associated environmental impact of their utilization, an increasing interest has been shifted towards the exploration of alternative and greener carbon sources[1]. Given its sustainable nature and global abundance, biomass has been widely acknowledged as the most promising carbon source whose utilization will not alter the current carbon balance of our ecosystem. However, the complex and often recalcitrant structure of biomass components necessitates the development of effective transformation strategies, namely biomass valorization[2]. Even though valuable products, such as polymer precursors (e.g., 2,5-furandicarboxylic acid[3], 2,5-bis (hydroxymethyl)furan[4], biofuels and additives (e.g., 2-methylfuran, 2,5-dimethyl furan)[5,6], and green solvents (e.g., γ-valerolactone)[7], have been obtained from biomass-derived furanics, the economic competitiveness of these bio-products against their fossil-derived counterparts is less well established if large-scale employment has yet been realized[8]. In contrast, because of the much higher value of pharmaceuticals per kilogram, even small-scale employment could be economically attractive and profitable if biologically active products could be synthesized from biomass-derived compounds. Nevertheless, there have been very few reports on biomass upgrading for medicinal applications. Against this backdrop, it emerges as an exciting direction to synthesize heterocyclic molecules from biomass-derived intermediate compounds, because heterocyclic motifs play pivotal roles in constructing biologically active products while biomass consists of many heterocyclic structural units[9,10].

Last few decades have witnessed the success of various synthetic strategies in creating heterocyclic units for medicinal applications, including hetero-Diels-Alder condensation[11,12], oxa-Michael addition[13], radical cyclization[14], and ring-closing olefin metathesis[15]. Nevertheless, most of these approaches require petrochemical-based starting materials, expensive reagents, and/or harsh conditions. Consequently, it is highly desirable to explore the potential of heterocyclic synthesis starting from biomass-derived compounds under ambient conditions. Within this context, Achmatowicz reaction stands out as an ideal candidate since its starting compound is furfuryl alcohol, which can be produced from a one-step reduction of furfural[16]. It should be noted that furfural is readily available from hemicellulose refinery[17]. Different from 5-(hydroxymethyl)furfural, furfural has an industrial scale production with a global yield close to 0.3 million tons per year[18].

The Achmatowicz reaction is able to convert furfuryl alcohol into substituted dihydropyranone acetals, which find applications in the synthesis of natural products and pharmaceuticals[19]. In addition, a chiral center is introduced through the Achmatowicz reaction, therefore many asymmetric variants could be developed subsequently, paving the routes to synthesize diastereoselective and optically active compounds[20]. Figure 1a presents a few representative biologically active products whose syntheses are benefited from the Achmatowicz reaction[21–23]. The Achmatowicz rearrangement was first reported in 1970s, utilizing $Br_2$ as the oxidant and methanol as the solvent to produce stable methoxylated furfuryl alcohol (Fig. 1b)[16]. Upon the treatment with $H^+$, hydropyranone was formed through rearrangement (Table 1, Entry 1). Later, it was demonstrated that N-bromosuccinimide (NBS) could be used for one-step production of hydropyranone, albeit at lower temperature (Table 1, Entry 2)[24]. In addition, the Achmatowicz reaction can also be driven by meta-chloroperoxybenzoic acid (m-CPBA)[25] via epoxidation at the olefin moiety of the furan ring in furfuryl alcohol (Table 1, Entry 3). Other oxygen-atom-transfer reagents including dimethyldioxiran were also effective (Table 1, Entry 4)[26]. More recently, the study of biocatalytic Achmatowicz reaction employing mono-oxygenase further expanded the reaction scope (Table 1, Entry 5)[27]. Nevertheless, the need for stoichiometric oxidants and organic solvents, many of which are either toxic or expensive, and/or complicated procedure, represents an apparent limitation of these chemical oxidation approaches.

In contrast, electrochemical synthesis is an appealing strategy in expanding the available methodology of the Achmatowicz reaction (Fig. 1c). In fact, electrochemical methoxylation of furfuryl alcohol was reported in 1950s using Pt and Ni as the anode

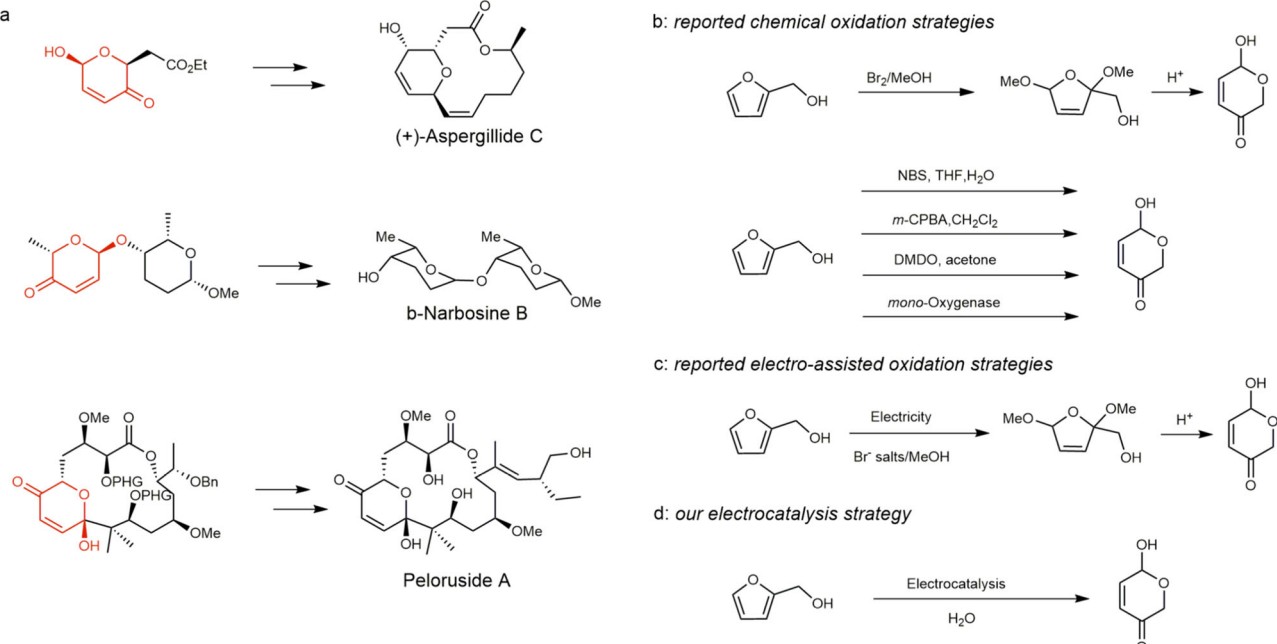

**Fig. 1 Application and synthesis development of hydropyranones. a** Representative compounds containing hydropyranone units used in the syntheses of biologically active compounds. **b** Reported chemical and **c** electricity-assisted oxidation strategies for the Achmatowicz reaction. **d** Our proposed electrocatalytic Achmatowicz reaction operating in water.

**Table 1 Summary of reported chemical oxidation strategies.**

| Entry | Oxidants | Condition | Post treatment | Yield (%) | Ref. |
|---|---|---|---|---|---|
| 1 | Br$_2$ | r.t. in MeOH | Hydrolysis with H$^+$ | 73 | 16 |
| 2 | NBS | 0 °C in THF/H$_2$O = 4:1 | –[a] | 65 | 24 |
| 3 | *m*-CPBA | r.t. in CH$_2$Cl$_2$ | – | 48 | 25 |
| 4 | DMDO | r.t. in acetone | – | 95 | 26 |
| 5 | *mono*-oxygenase | D-glucose, and glucose oxidase in citrate buffer/t-butanol = 9:1 | L-methionine was added before extraction | 82 | 27 |

[a]No post-treatment needed.

and cathode, respectively, achieving a yield of 66% with a faradaic efficiency of 77% (Table 2, Entry 1)[28,29]. The reaction was carried out in MeOH with the presence of NH$_4$Br at −15 °C using a fairly high voltage of 4.8 V. Hydrolysis of the methoxylated furfuryl alcohol was required to produce hydropyranone[30]. This method was later modified slightly to proceed at 0 °C with comparable yield and faradaic efficiency (Table 2, Entry 2)[31]. A similar electrochemical strategy was reported employing immobilized furfuryl alcohol on the electrode to prepare the methoxylated furfuryl alcohol derivatives followed by H$^+$ treatment to complete the rearrangement (Table 2, Entry 3)[32]. The final hydropyranone derivatives were then liberated from the electrode after post-treatment. Even though the product yield was comparable to the previous reports, its faradaic efficiency was as low as 5%. A common observation of these electrochemical approaches is the requirement of bromide salts. Even though these strategies have been adopted in a number of publications and even industrial application[33–36], there is no breakthrough for decades. Alternatively, direct methoxylation of furanics through the furanoxonium ion pathway requires substantially high voltage (Table 2, Entry 4)[37]. Overall, the requirement of bromide salts, organic solvents (e.g., MeOH), low temperature, high voltage, and additional post-treatment are apparent limitations of these reported electrochemical strategies for the Achmatowicz reaction from both economic and environmental perspectives. It would be more desirable to develop an electrocatalytic approach operating under ambient conditions and preferably using water as the oxygen source as well as solvent (Fig. 1d).

Herein, we report an electrocatalytic strategy for the Achmatowicz reaction in aqueous media, using water as the sole oxygen source (Table 2, Entry 5). Taking advantage of the photo-induced immobilization of a bipyridyl ligand, 5,5'-divinyl-2,2'-bipyridine (dvbpy), on conductive substrate, we are able to incorporate nickel sites with unsaturated coordination spheres, which enable the intimate interaction between furfuryl alcohol and nickel and in turn to allow the Achmatowicz reaction to take place. Our findings enrich the arsenal of organic chemists and provide green and economically attractive methods in synthesizing heterocyclic molecules, which can find unique applications in constructing biologically active molecules and pharmaceuticals.

## Results and discussion

**Catalyst preparation and characterization.** To take the advantages of both homogeneous (defined active sites at the molecular level)[38] and heterogeneous (efficient electron transfer between electrode and catalyst without diffusion) electrocatalysts, we immobilized molecular active sites on electrode for the Achmatowicz reaction. Indeed, many elegant approaches, such as ππ stacking[39], covalent tethering[40,41], and polymerization[42], have been adopted for catalyst immobilization. However, most of these precedent procedures utilize pre-synthesized molecules and their active sites are usually coordination saturated. Herein, we

introduce an alternative strategy, pre-immobilization of a ligand film followed by metal cation incorporation (Fig. 2), enabling the facile access of varying electrocatalysts with unsaturated coordination spheres, which are not easily accessible in homogeneous synthesis.

As illustrated in Fig. 2, 5,5'-divinyl-2,2'-bipyridine (dvbpy) was first drop casted on a conductive substrate. Subsequent UV light irradiation ($\lambda_{irr} = 250$ nm) initiated its polymerization and resulted in a ligand film (DVBP) uniformly covered on the substrate (see Methods section for details). Such a facile photo-induced polymerization can be applied to prepare DVBP on a variety of solid substrates, including glassy carbon and carbon paper with high surface area. Figure 3a compares the Fourier-transform infrared (FT-IR) spectra of dvbpy and DVBP, confirming the retention of the characteristic vibrational features of the 2,2'-bipyridyl structure in DVBP. The decreased ratio of the C=C/C=N stretching band at 1633 and 1589 cm$^{-1}$ after polymerization compared with the pristine dvbpy is consistent with polymerization through the vinyl groups[43,44]. Direct immersing DVBP in an acetonitrile solution of nickel triflate produced the desirable nickel catalyst Ni-DVBP. The comparison of the Raman spectra of DVBP and Ni-DVBP is presented in Fig. 3b, wherein all the distinctive peaks in the region of 1100–1700 cm$^{-1}$ owing to the parent ligand film still remained after Ni incorporation. It was worth noting that the Raman spectrum of Ni-DVBP presented a new peak at 396 cm$^{-1}$, which could be assigned to the Ni-N stretching[45]. Scanning electron microscopy (SEM) images of Ni-DVBP at different magnifications (Figs. 3c and 3d) imply its rough and porous morphology, which is anticipated to be beneficial for electrolyte penetration and the accessibility of active sites during electrocatalysis. Energy-dispersive X-ray spectroscopy data further proved the presence of both Ni and N in Ni-DVBP (Supplementary Fig. 2). Finally, X-ray photoelectron spectroscopy (XPS) was conducted to probe the composition and valence state of each element in Ni-DVBP. As shown in Fig. 3e, the high-resolution Ni 2$p$ XPS spectrum can be deconvoluted to four subpeaks at binding energies of 855.7, 873.2, 861.6, and 880.3 eV, assignable to Ni$^{2+}$ 2$p_{3/2}$, 2$p_{1/2}$, and two corresponding satellite peaks, respectively. Fig. 3f presents the high-resolution N 1$s$ XPS spectrum and the observed peak at 399.2 eV could be ascribed to an anticipated pyridine N in Ni-DVBP, in agreement with the previous reports[46].

The electrochemical behavior of Ni-DVBP prepared on a glassy carbon electrode was first investigated in acetonitrile. As implied by its cyclic voltammogram (CV, Fig. 4a) and square wave voltammogram (Supplementary Fig. 3), a pseudo reversible feature at −0.31 V vs Fc$^{+/0}$ was observed, which could be assigned to the Ni$^{3+/2+}$ redox couple. The linear relationship of the anodic and cathodic maxima of this redox feature versus scan rate confirmed the immobilization nature of the nickel sites on the electrode rather than free diffusion in the electrolyte (Fig. 4b). DVBP and Ni-DVBP could be prepared on carbon paper in an analogous manner as on glassy carbon. Owing to the high surface

**Table 2 Comparison of reported electro-assisted oxidation strategies and our electrocatalysis strategy.**

| Entry | Electrode (anode/cathode) | Electrolyte | Potential/current | Yield (%) | Post treatment | Faradaic efficiency (%) | Ref. |
|---|---|---|---|---|---|---|---|
| 1 | Pt/Ni | 0.2 M NH$_4$Br in MeOH (−15 °C) | 4.8 V/2.5 A | 66[a] | Hydrolysis with H$^+$ | 77 | 28–30 |
| 2 | C/Cu | 0.2 M NaBr in EtOH (0 °C) | N/A | 67[a] | Hydrolysis with H$^+$ | 80 | 31 |
| 3 | C/C | 0.2 M Bu$_4$NBr in MeOH/dioxane = 1:1 (0 °C) | 40 mA | 63[a] | Hydrolysis with H$^+$ and deimmobilization | 5 | 32 |
| 4 | GC/Pt | 0.05 mM NaClO$_4$ in MeOH | 3.9 V/14.6 mA | 80[a] | Hydrolysis with H$^+$ | 60 | 37 |
| 5 | Ni-DVBP/Pt | 0.1 M phosphate buffer (r.t.) | 1.4 V vs Ag/AgCl | 96[b] | –[c] | 84 | This work |

[a] The yield of methoxylated furfuryl alcohols.
[b] The yield of hydropyranone.
[c] No post-treatment needed.

area of carbon paper, which is beneficial for electrocatalysis, the following mentioned DVBP and Ni-DVBP samples were all prepared on carbon paper unless noted otherwise. In fact, an apparent color change from gray to black could be observed after the coating of DVBP on carbon paper (Supplementary Fig. 4a). SEM images of carbon paper prior to (Supplementary Fig. 4b) and post (Supplementary Fig. 4c) DVBP coating clearly showed that its originally smooth carbon fibers were covered by a web-like ligand film. The pristine carbon paper exhibited an appreciable capacitance current within 0.4 and 0.8 V vs Ag/AgCl in 0.1 M phosphate buffer (KPi) at pH 7 (Fig. 4c). Once DVBP was coated, a slightly higher capacitance current was observed in the same potential region. Such a large capacitance current obscured the observation of the Ni$^{3+/2+}$ redox feature in Ni-DVBP prepared on carbon paper. Nevertheless, the much smaller onset potential of O$_2$ evolution on Ni-DVBP compared to those on the pristine carbon paper and DVBP unambiguously supported the incorporation of Ni$^{2+}$ in the prepared Ni-DVBP. For the sake of clarity, the CV curve of Ni-DVBP was replotted in Fig. 4d and overlaid with the CV trace after the addition of 10 mM furfuryl alcohol. A much rapid anodic current take-off at a smaller onset potential (~0.9 V vs Ag/AgCl) strongly suggests the oxidation of furfuryl alcohol on Ni-DVBP. It is necessary to mention that carbon paper and DVBP required more positive potential to oxidize furfuryl alcohol and the obtained anodic currents were also much less pronounced relative to the case on Ni-DVBP (Supplementary Fig. 5).

**Electrolysis conditions development**. As furfuryl alcohol could be oxidized to furfural or follow the Achmatowicz reaction to yield hydropyranone, it was critical to identify and quantify the oxidation products obtained on Ni-DVBP. Bulk electrolysis in a three-electrode configuration using Ni-DVBP as the working electrode, Ag/AgCl as the reference electrode, and carbon rod as the counter electrode in a two-compartment cell was carried out (Method A, see Supplementary Information for details). As shown in Table 3 (entry 1), electrolysis conducted at 1.4 V vs Ag/AgCl achieved a high yield (96%) of hydropyranone together with a superior selectivity over furfural, whose yield was merely 2%. Furthermore, the calculated faradaic efficiency was 84% with the loading of 10 mM furfuryl alcohol. In fact, if the starting concentration of furfuryl alcohol increased to 50 mM, a much higher faradaic efficiency of 93% could be obtained, indicating that furfuryl alcohol oxidation was preferred over O$_2$ evolution as long as there was sufficient amount of furfuryl alcohol in the electrolyte. If only DVBP (entry 2) or a pristine carbon paper (entry 2) was used as the working electrode under the same condition, even though hydropyranone could be detected, its yield, selectivity, and faradaic efficiency were all much inferior to those obtained on Ni-DVBP, strongly implying the unique catalytic activity of Ni-DVBP in driving the Achmatowicz reaction. The formation of hydropyranone on carbon paper and DVBP could be owing to the direct oxidation of furfuryl alcohol through an epoxidation pathway, which will be discussed later. In a 0.1 M acetate buffer of pH 5, Ni-DVBP was also able to catalyze the formation of hydropyranone with a decent yield of 92%, but a mediocre far-adaic efficiency of 63% was obtained (entry 4). On the other hand, alkaline condition favors furfural generation, so negligible hydropyranone was detected on Ni-DVBP at pH 10 (entry 5). These results demonstrate that it is feasible to readily tune the electrolyte pH to selectively produce hydropyranone or furfural using the same electrocatalyst Ni-DVBP.

High-performance liquid chromatography was utilized to monitor the concentration evolution of furfuryl alcohol and its oxidation products on Ni-DVBP during electrolysis at 1.4 V vs

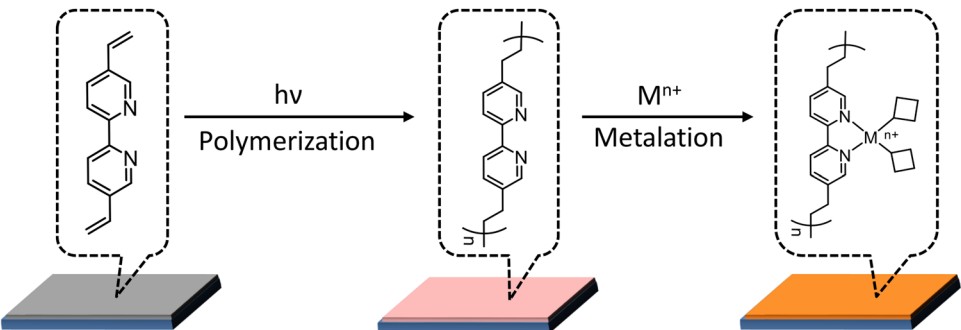

**Fig. 2 Illustration of catalyst preparation.** Photo-induced polymerization of dvbpy followed by metal cation incorporation to prepare M-DVBP electrocatalysts.

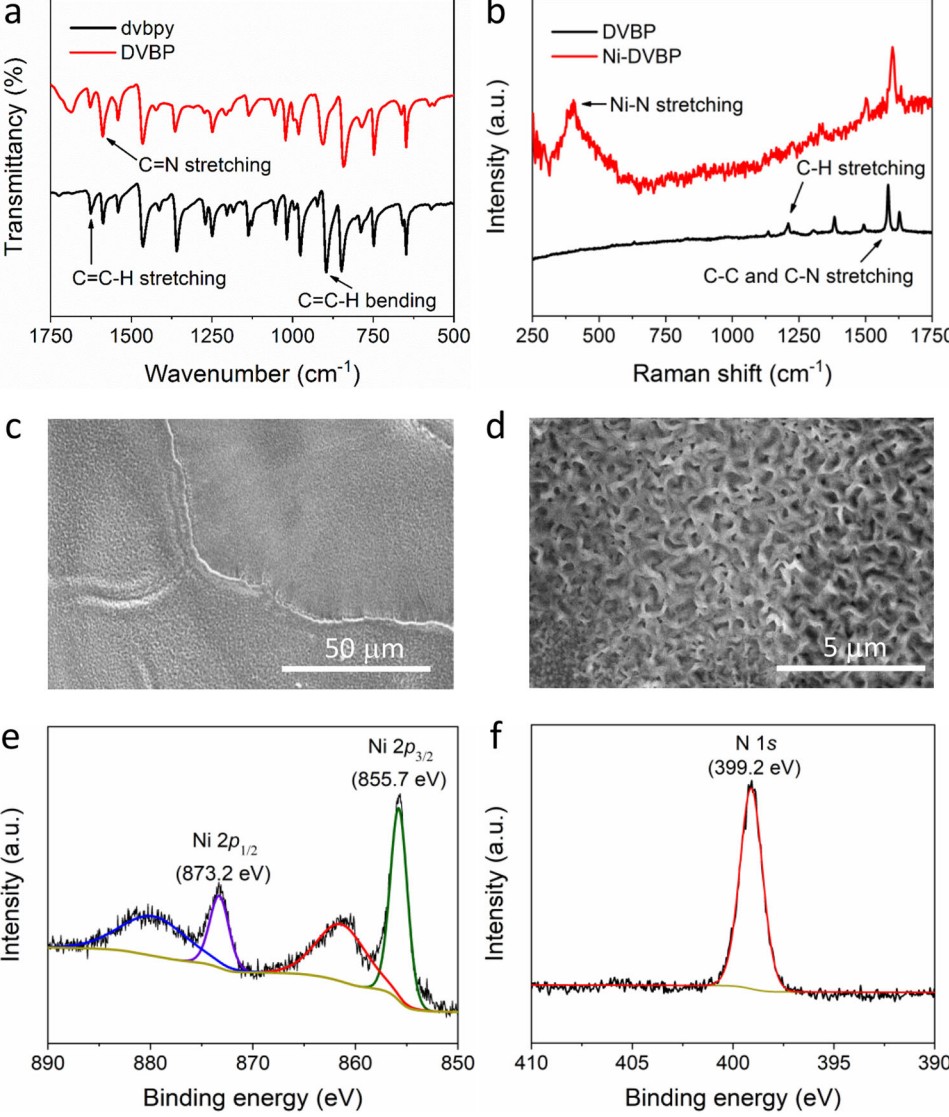

**Fig. 3 Catalyst characterization. a** FT-IR spectra of dvbpy and DVBP. **b** Raman spectra of DVBP and Ni-DVBP. **c, d** SEM images of Ni-DVBP at different magnifications. High-resolution XPS spectra of Ni 2$p$ **e** and N 1$s$ **f**.

Ag/AgCl in the 0.1 M phosphate buffer of pH 7 (Method A, see Supplementary Information for details). As shown in Fig. 5a, hydropyranone emerged at the onset of electrolysis and its concentration proportionally increased along the passed charge. In contrast, the yield of furfural remained very low (<2%) throughout the entire course of electrolysis. Four successive cycles

of electrolysis using the same Ni-DVBP electrode were performed. As plotted in Fig. 5b, the nearly complete furfuryl alcohol conversion and almost identical high yield of hydropyranone unambiguously proved the superior robustness of our Ni-DVBP for the electrocatalytic Achmatowicz reaction. Following Method B (see Supplementary Information for details), hydropyranone could also

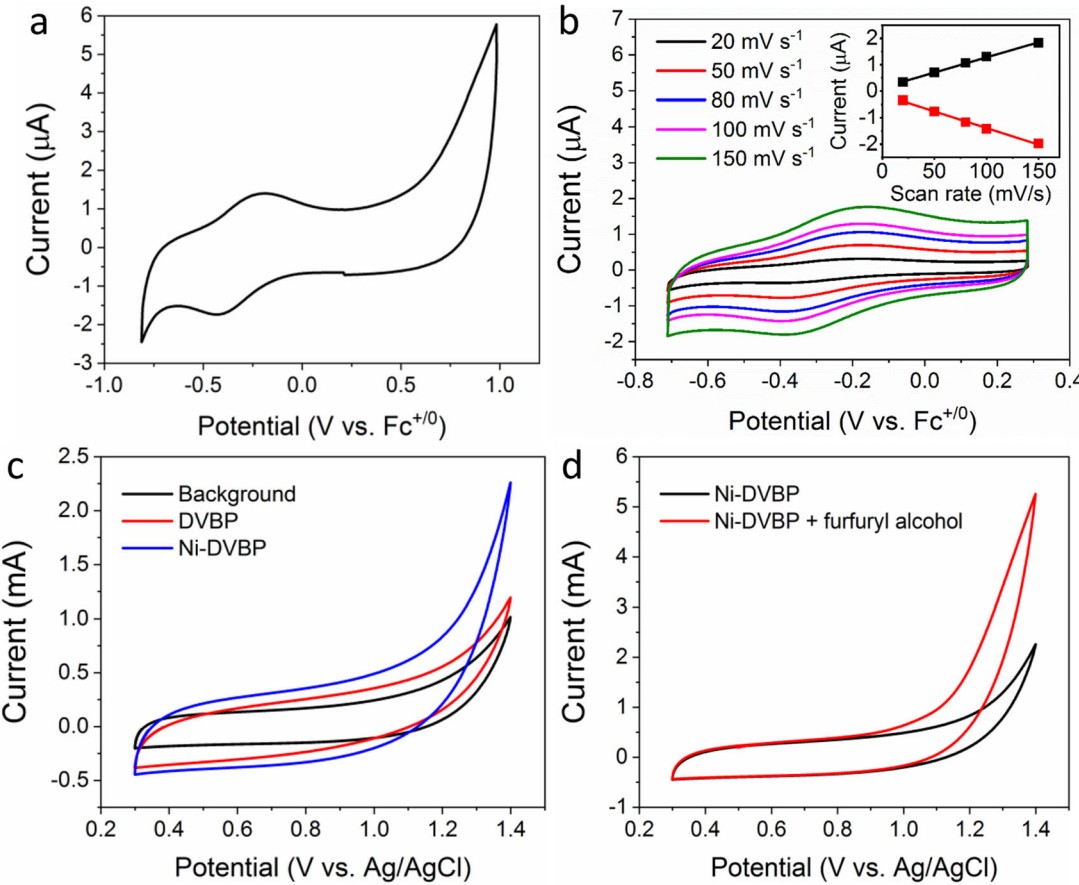

**Fig. 4 Electrochemistry study of Ni-DVBP. a** CV curve of Ni-DVBP collected in acetonitrile at a scan rate of 100 mV/s. **b** Scatter plot of anodic and cathodic peak currents of Ni-DVBP versus scan rate, together with linear fitting lines. **c** CV comparison of carbon paper, DVBP, and Ni-DVBP collected in 0.1 M KPi at pH 7.0. **d** CV curves of Ni-DVBP with and without 10 mM furfuryl alcohol in 0.1 M KPi at pH 7.

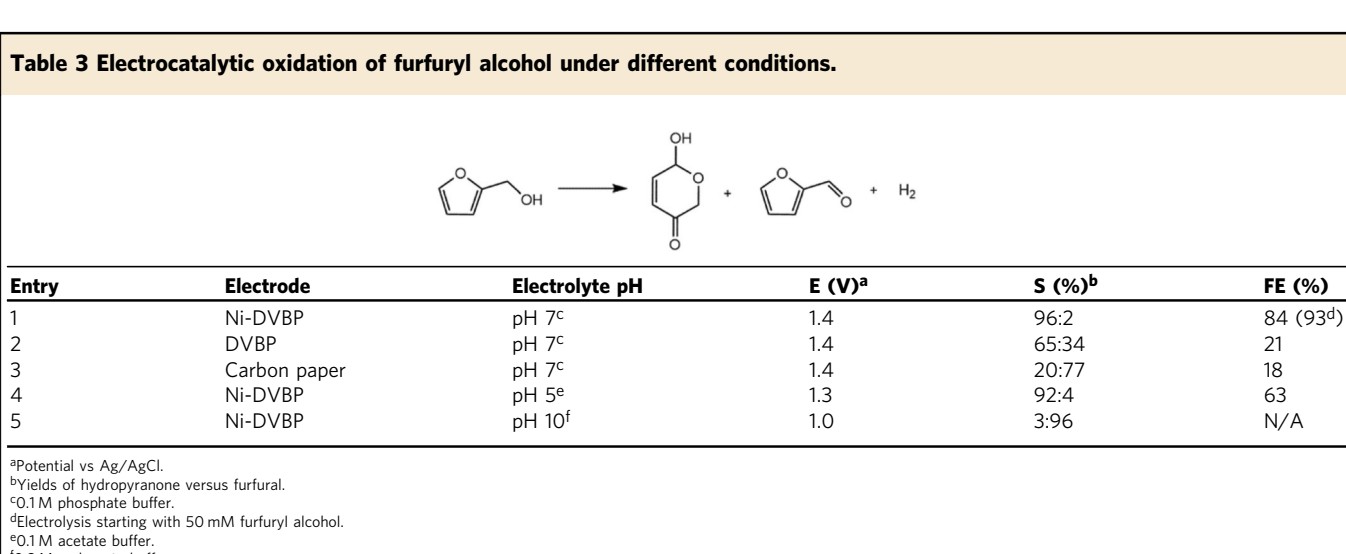

**Table 3 Electrocatalytic oxidation of furfuryl alcohol under different conditions.**

| Entry | Electrode | Electrolyte pH | E (V)[a] | S (%)[b] | FE (%) |
|---|---|---|---|---|---|
| 1 | Ni-DVBP | pH 7[c] | 1.4 | 96:2 | 84 (93[d]) |
| 2 | DVBP | pH 7[c] | 1.4 | 65:34 | 21 |
| 3 | Carbon paper | pH 7[c] | 1.4 | 20:77 | 18 |
| 4 | Ni-DVBP | pH 5[e] | 1.3 | 92:4 | 63 |
| 5 | Ni-DVBP | pH 10[f] | 1.0 | 3:96 | N/A |

[a]Potential vs Ag/AgCl.
[b]Yields of hydropyranone versus furfural.
[c]0.1 M phosphate buffer.
[d]Electrolysis starting with 50 mM furfuryl alcohol.
[e]0.1 M acetate buffer.
[f]0.2 M carbonate buffer.

be isolated with a yield of 92%. To demonstrate the feasibility of our electrocatalytic approach for the large-scale synthesis of hydropyranone, a flow electrolyzer was utilized (Method C). A gram scale electrolysis was performed and pure hydropyranone was obtained with an isolated yield of 84% after simple workup. As a comparison, two chemical oxidation approaches using NBS and *m*-CPBA as the oxidants were employed to synthesize hydropyranone, resulting in the hydropyranone yields of 73% and 82%, respectively, which were comparable with reported yields[47,48].

**Substrate scope**. The success of the Achmatowicz reaction electrochemically catalyzed by Ni-DVBP prompted us to explore its versatility towards other furfuryl alcohol derivatives. In particular,

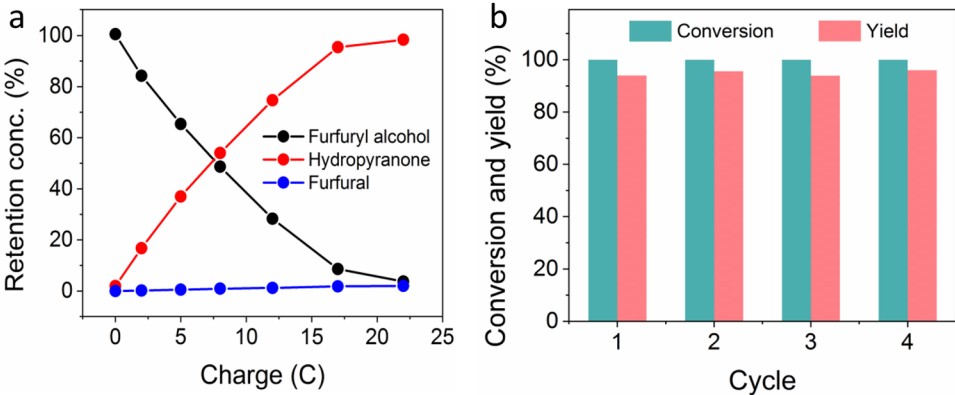

**Fig. 5 Quantification of the electrocatalytic Achmatowicz reaction. a** Conversion of furfuryl alcohol and yields of its oxidation products over passed charge during electrolysis at 1.4 V vs Ag/AgCl in 0.1 M phosphate buffer using Ni-DVBP as the working electrode. **b** Conversion of furfuryl alcohol and yield of hydropyranone for four consecutive electrolysis cycles at 1.4 V vs Ag/AgCl in 0.1 M phosphate buffer utilizing the same Ni-DVBP as the working electrode.

**Fig. 6 Substrate scope.** The Achmatowicz reaction yields of furfuryl alcohol derivatives obtained on Ni-DVBP at 1.4 V vs Ag/AgCl in 0.1 M phosphate buffer of pH 7. The corresponding product yields obtained from chemical oxidation using *m*-CPBA as the oxidant are also included for comparison. *Electrolysis at 1.8 V vs Ag/AgCl.

we focused on those derivatives with various substituents positioned at the hydroxymethyl position and the 5'-position of the furan ring, in that these positions are crucial for the success of the Achmatowicz reaction to take place. As shown in Fig. 6, a methyl

substituent at the hydroxymethyl position (**1b**) did not prohibit the formation of **2b** and a 93% yield was obtained. Introducing more sterically bulky substituents like dimethyl (**1c**), propyl (**1d**), benzyl (**1e**), and phenethyl (**1f**) groups at the same position

resulted in comparable yields as that obtained from the pristine furfuryl alcohol. As a comparison, when *m*-CPBA was utilized as the chemical oxidant, the conventional Achmatowicz reactions were also conducted for the aforementioned furfuryl alcohols and the corresponding yields of hydropyanones are included in Fig. 6 as well. It is apparent that our electrocatalytic Achmatowicz strategy produced comparable or even better yields than those from *m*-CPBA driven reactions. Next, the electronic effect was also investigated by evaluating substrates **1g**, **1h**, and **1i**. The more electron-withdrawing *p*-fluorophenyl group (**1i**) resulted in a lower yield (76%) of **2i** compared with the others (~90%). However, the hydropyranone yields (66–73%) obtained from *m*-CPBA-driven reactions did not show apparent difference among these three substrates, implying a different mechanism from our electrocatalytic Achmatowicz approach.

Furfuryl alcohol derivatives with substituents at the 5'-position of the furan ring were also studied. A methyl group at the 5'-position in **1j** resulted in a slightly lower yield (82%) of 2j compared with that of **2b** (93%), probably due to the steric hindrance. A more bulky substituent like *p*-chlorophenyl at the 5'-position (**1k**) would completely suppress the electrocatalytic Achmatowicz reaction conducted at 1.4 V vs Ag/AgCl. Nevertheless, following the oxygen-atom transfer mechanism, the steric hindrance did not prevent the formation of **2j** and **2k** when *m*-CPBA was used as the chemical oxidant. The mechanistic steps of the electrocatalytic Achmatowicz reaction on Ni-DVBP versus driven by oxygen-atom-transfer reagents like *m*-CPBA will be further discussed later.

To further demonstrate the applicability of our electrocatalytic Achmatowicz strategy towards highly valuable reactions, we evaluated furfuryl alcohol derivatives with unique functional groups that are essential in the synthesis of natural products. For example, the Achmatowicz reaction has been adopted o synthesize aspergillide and its analogs[49]. The synthesis of **2l** is critical to introduce the six-member ring and its ester group plays a vital role to the ring-closure step in the synthesis of the final product (-)-Aspergillide A. Our electrocatalytic Achmatowicz strategy was successfully applied in the synthesis of **2l** and the yield was comparable to that using *m*-CPBA as the chemical oxidant. In addition, an oxidation sensitive vinyl group was well tolerated in our electrocatalytic Achmatowicz reaction method in the preparation of **2m**, paving the way for the synthesis of halichondrins[50]. These two successful examples further showcase the promise of our electrocatalytic Achmatowicz reaction as a green and facile approach for practical and useful organic transformations.

**Mechanistic study**. To aid the understanding of the electrocatalytic Achmatowicz reaction on Ni-DVBP, density functional theory (DFT) calculations at the B3LYP/6-31 G(d) level were performed. Based on the FT-IR and XPS data, we proposed a molecular model of a mononuclear Ni coordinated with one 2,2'-bipyridyl and two aqua ligands, $[(bpy)Ni^{II}(H_2O)_2]^{2+}$, to mimic the resting state of the Ni active sites in Ni-DVBP. With $[(bpy)Ni^{II}(H_2O)_2]^{2+}$ as the starting model, our DFT calculation results indicate that it is thermodynamically favorable for furfuryl alcohol to replace one of the aqua ligands with an energy decrease of −0.21 eV (intermediate **2** shown in Fig. 7).

The interaction between furfuryl alcohol and Ni-DVBP was investigated by comparing the corresponding IR spectra of Ni-DVBP, furfuryl alcohol, and Ni-DVBP treated with furfuryl alcohol (Ni-DVBP + furfuryl alcohol) (Supplementary Fig. 6). Specifically, Ni-DVBP was immersed in an aqueous solution of furfuryl alcohol overnight and subsequently rinsed thoroughly with water. Next, it was dipped in water under stirring for 5 h to remove any loosely adsorbed furfuryl alcohol and finally Ni-

DVBP + furfuryl alcohol was obtained for IR measurement. As shown in Supplementary Fig. 6b, the major IR features of free furfuryl alcohol at 811.4, 909.9, 1002.5, and 1189.2 $cm^{-1}$ could be ascribed to its HC2C3H rocking, C1O stretching, C5O bending, and C1C5 stretching, according to the literature assignments[51]. In the IR spectrum of Ni-DVBP + furfuryl alcohol (Supplementary Fig. 6c), these furfuryl alcohol features also appeared but were shifted to 794.4, 922.2, 1012.6, and 1168 $cm^{-1}$, respectively. In addition, the pyridine C = N bending peak of Ni-DVBP was also shifted from 1464.0 $cm^{-1}$ (Supplementary Fig. 6a) in Ni-DVBP to 1475.4 $cm^{-1}$ (Supplementary Fig. 6c) in Ni-DVBP + furfuryl alcohol[52]. Overall, these IR spectra demonstrate that furfuryl alcohol is indeed bound within Ni-DVBP and the mutual influence of their IR features strongly supports the ligation of furfuryl alcohol to the Ni center in Ni-DVBP, in agreement with the above DFT calculated formation of intermediate **2**.

DFT calculation was also performed to probe the subsequent oxidation and deprotonation of intermediate **2**, which triggered the interaction between the $Ni^{II}$-OH center and the 5'-position carbon in the bound furfuryl alcohol as shown as intermediate **3** in Fig. 7. This step requires an energy input of 0.79 eV. Followed by the coordination of another solvent water molecule to Ni in **4**, the hydroxide group originally ligated on Ni migrates to the 5' position of furfuryl alcohol whose alcohol group is liberated from the Ni center. This step is energetically favorable since the energy of **4** is 0.37 eV lower than that of **3**. The second oxidation and deprotonation occur to **4** would further drive the rearrangement process (Supplementary Fig. 7) to liberate intermediate **5** and back to the original state of $[(bpy)Ni(H_2O)_2]^{2+}$ after accepting a solvent water molecule. This step is very energetically downhill from 0.21 eV to −0.60 eV. It should be noted that intermediate **5** is a well-anticipated intermediate in the Achmatowicz reaction, which will undergo the final ring-closure step to furnish the formation of hydropyranone.

We also considered the scenario where there is no direct interaction between the Ni center and furfuryl alcohol (Mechanism 2 in Fig. 7). In this case, $[(bpy)Ni^{II}(H_2O)_2]^{2+}$ would undergo two consecutive oxidation processes with applied potential to generate intermediate **3'**, which possesses a Ni-oxyhydroxide moiety. These two oxidation steps need high energy inputs of 2.62 and 2.02 eV, respectively, which are substantially higher compared to those in Mechanism 1. An oxygen-atom transfer could take place between Ni-oxyhydroxide and furfuryl alcohol to release $[(bpy)Ni^{II}(H_2O)_2]^{2+}$ and intermediate **4'**, followed by the formation of the rearrangement product hydropyranone. It should be noted that Ni-oxyhydroxide has been widely accepted as the active species towards $O_2$ evolution in neutral and alkaline electrolytes[53], a competing reaction of furfuryl alcohol oxidation. In short, our DFT calculation results indicate that Mechanism 2 is more energetically demanding than Mechanism 1 (Supplementary Fig. 8) and even if Mechanism 2 takes place, the competing $O_2$ evolution reaction is unavoidable.

Given the calculated difference in energy requirement between Mechanisms 1 and 2, we reasoned that the preferred oxidation pathway of the electrocatalytic Achmatowicz reaction would be highly dependent on applied potential. Therefore, a series of electrocatalytic oxidation of furfuryl alcohol on Ni-DVBP and blank carbon paper were conducted under the same condition at 1.1, 1.2, 1.3, and 1.4 V vs Ag/AgCl (Supplementary Fig. 9). Supplementary Fig. 10 and Table S1 present the corresponding yields of hydropyranone obtained on Ni-DVBP and carbon paper. As indicated in Fig. 3d, 1.1 V vs Ag/AgCl is the onset potential of furfuryl alcohol oxidation on Ni-DVBP. Nearly 100% conversion of furfuryl alcohol and decent hydropyranone yields of 79% and 87% were obtained on Ni-DVBP at 1.1 and 1.2 V, respectively. In sharp contrast, electrolysis conducted on blank

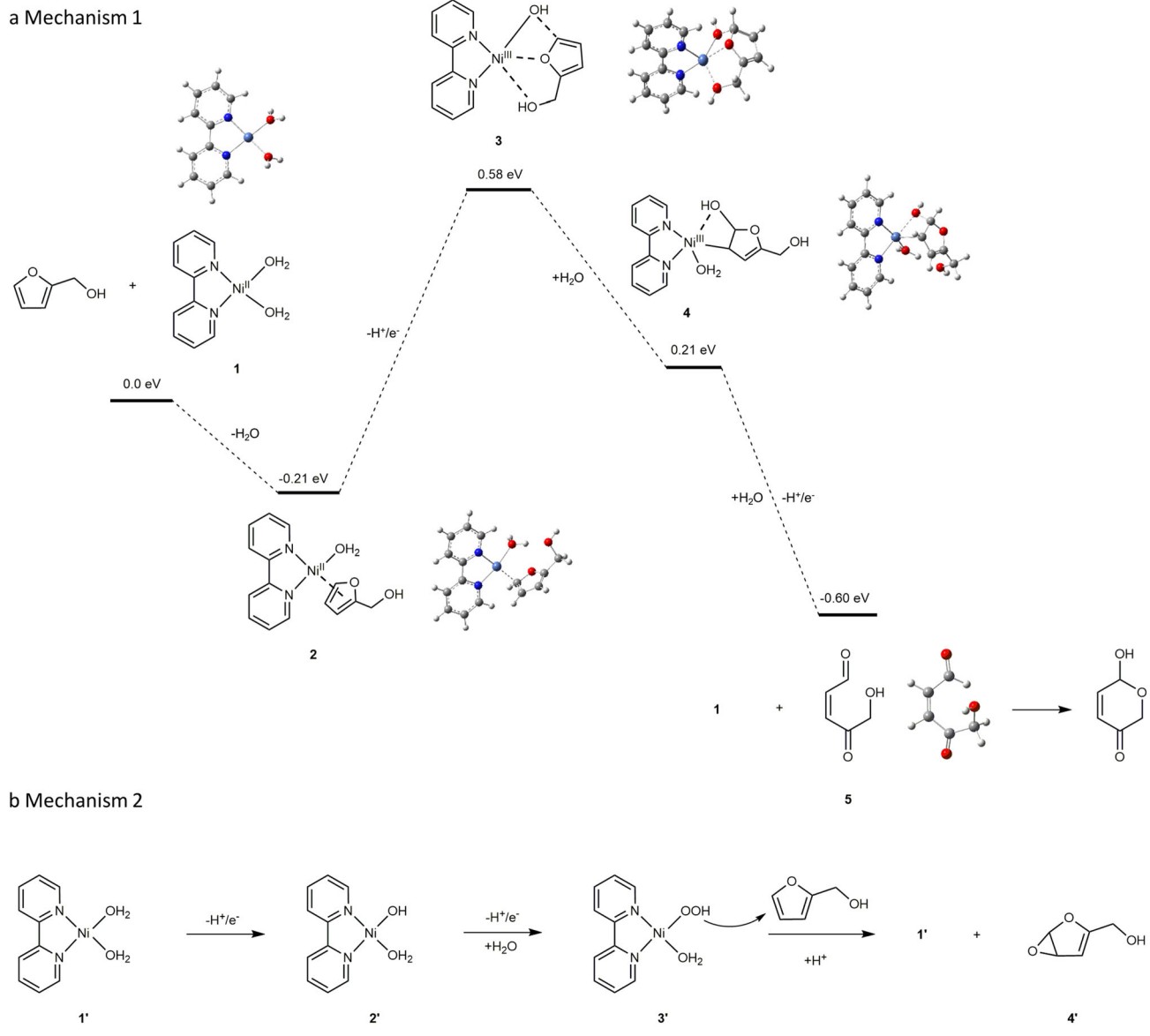

**Fig. 7 Plausible reaction mechanisms. a** Mechanism 1: DFT calculated energy profile of catalytic intermediates using [(bpy)Ni(H$_2$O)$_2$]$^{2+}$ as a catalyst model. The optimized chemical structure of each intermediate is also included. **b** Mechanism 2: The chemical structure of each intermediate using the same catalyst model but following the oxygen-atom transfer pathway.

carbon paper resulted in negligible furfuryl alcohol conversion and hydropyranone yield at these applied potentials. Along the increase of applied potential to 1.3 and 1.4 V, the resulting hydropyranone yields reached 92% and 96%, respectively, on Ni-DVBP. Because carbon paper started to exhibit water oxidation capability beyond 1.3 V vs Ag/AgCl (Fig. 3c), it is reasonable to anticipate that furfuryl alcohol oxidation could also take place on carbon paper at 1.3 and 1.4 V. Indeed, a small amount of hydropyranone (<20%) was detected when carbon paper was employed as the working electrode. Further increasing the applied potential to 1.8 V vs Ag/AgCl, wherein vigorous O$_2$ evolution could be observed on both Ni-DVBP and carbon paper, the addition of furfuryl alcohol led to the formation of hydropyranone with 93% and 65% yields on Ni-DVBP and carbon paper, respectively, albeit accompanied by much lower Faradaic efficiencies (67% on Ni-DVBP and 53% on carbon paper). At such a high applied potential, the calculated Mechanism 2 could be viable, even though the same active species (e.g., Ni-oxyhydroxide) is also very active for O$_2$ evolution.

Encouraged by the above potential-dependent electrolysis results of Ni-DVBP, we went back to reinvestigate the oxidation of substrate **1k** whose *p*-chlorophenyl substituent at the 5'-position prohibited its oxidation on Ni-DVBP at 1.4 V vs Ag/AgCl. When the applied potential was 1.8 V vs Ag/AgCl, a high yield (84%) of the corresponding hydropyranone product **2k** was obtained. Such a drastic difference in the oxidation of substrate **1k** at 1.4 and 1.8 V indicate that two distinct oxidation mechanisms are involved. At low applied potentials (1.1–1.4 V Ag/AgCl), intramolecular hydroxide transfer from the nickel center to the 5'-position of furfuryl alcohol is the dominant pathway, which is strongly influenced by the bulky substituents at the 5'-position. However, at high applied potential (e.g., 1.8 V vs Ag/AgCl), direct oxygen-atom transfer likely takes place between Ni-DVBP (and electrode) and furfuryl alcohol. Nevertheless, much lower Faradaic efficiency is obtained because of the severe competition of the O$_2$ evolution process.

In summary, we report an electrocatalyst with molecular nickel active sites immobilized in a polymeric ligand film, which exhibits

superior selectivity, yield, faradaic efficiency, and robustness in driving the Achmatowicz reaction electrocatalytically. Such an electrocatalytic Achmatowicz reaction presented many advantages compared with its conventional synthetic approaches. First, a heterogenized Ni-DVBP electrocatalyst is adopted, rendering an environmentally benign reaction condition. Second, water is not only used as the reaction solvent but also the oxygen source. Toxic and/or expensive chemical oxidants are completely circumvented, hence no other byproducts or wastes are generated, resulting in easy separation and purification of the product and high atom economy. Finally, $H_2$ is produced at the counter electrode simultaneously, which is another valuable product. At the end, it should be noted that photo-induced ligand polymerization described herein provides a convenient method to prepare many other first-row transition metal-based electrocatalytic systems with unsaturated coordination spheres, which are hard to achieve under conventional homogeneous conditions, therefore numerous applications could be envisaged.

## Methods

**Preparation of catalyst**. Carbon paper was soaked with 0.5 M HCl to remove potential metal impurities and washed with copious water. In all, 20 mg of 5,5'-divinyl-2,2'-bipyridine was added in 0.5 mL isopropanol and 10% benzoyl peroxide as radical initiator to prepare stock solution. In all, 10 μL stock solution was added to one side of carbon paper (1 × 1 cm) and was placed under UV light (250 nm) for 15 min. The distance between light and carbon paper was 10 cm. Another 10 μL stock solution was added to the other side of carbon paper and was placed under UV light for 15 min to prepare DVBP. Subsequently, the as-prepared polymer was placed in a solution of 100 mM Ni(OTf)$_2$ in MeCN overnight and Ni-DVBP was obtained. Electrodes were washed with MeCN before use.

**Electrochemistry**. Electrochemical measurements were performed on a VMP-3 potentiostat (Biologic Science Instrument) with a three-electrode configuration. For Achmatowicz reaction, the as-prepared Ni-DVBP was directly used as the working electrode, a Ag/AgCl (sat. KCl) electrode as the reference electrode, and a carbon rod or platinum wire as the counter electrode. The experiments were conducted in 0.1 M phosphate buffer solution (pH 7.0) with and without 10 mM organic substrates in H-cell. Working and reference electrode were placed in working chamber and counter electrode was placed in counter chamber.

**Computational calculation details**. All the calculations were done with the Gaussian 09 program package[54] using the B3LYP hybrid functional[55,56]. The LANL2DZ basis set[57] with the effective core potential was employed for nickel with the 6-31 G(d, p) basis set for H, C, O, and N[58], which is further refined with the 6-311 + G(2df,2p) basis set after geometry optimization. Stationary points were found using the standard Berny optimization technique implemented in Gaussian 09 and confirmed via the vibrational analysis to make sure that minima show no imaginary frequency. The different spin states of the intermediate structures were compared to determine the most stable one. The reaction free energy of the electrochemical steps with the involvement of proton-electron pair were calculated using the computational hydrogen electrode approach[59]. The change in Gibbs free energy ($\Delta G$) for each reaction step is defined as $\Delta G = \Delta E + \Delta E_{ZPE} - T\Delta S$, where $\Delta E$ is the difference in the DFT total energy, $\Delta E_{ZPE}$ is the zero-point energy difference calculated from the vibrational frequencies, and $\Delta S$ is the entropy difference between the products and the reactants. Coordinates and spin states of all structures found are listed at the end of supporting information.

## Data availability

Detailed experimental procedures and characterization of compounds can be found in the Supplementary Information.

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

## Acknowledgements

Y.S. acknowledges the financial support of National Science Foundation (CHE1914546), Herman Frasch Foundation (820-HF17), and the University of Cincinnati. Z.C. is grateful for the financial support from the State Key Laboratory of Coal Conversion, Institute of Coal Chemistry, and Synfuels China, Co. Ltd. Z.C. also acknowledges the Hundred-Talent Program of Chinese Academy of Sciences and Shanxi Hundred-Talent Program (Y9SW911981). D.J. was sponsored by the US Department of Energy, Office of Science, Office of Basic Energy Sciences, Chemical Sciences, Geosciences, and Biosciences Division.

## Author contributions

Y.S. conceived and supervised the study. Xu.L. and Y.S. wrote the manuscript. Xu.L. and G.H. performed the experiments. B.L. and D.J. performed the DFT calculations. Xi.L. and Z.C. contributed to the catalyst characterization.

## Competing interests

The authors declare no competing interests.
