## [Peer Review File · Nature Communications]

REVIEWER COMMENTS

Reviewer #1 (Remarks to the Author):

The authors report the development of a novel electrocatalyst and its application to the formation of hydroxyranones from furfuryl alcohols. This represents, to my knowledge, the first example of a direct electrochemical Achmatowicz reaction, however the same overall transformation is easily achievable using either electrochemical (by post-treatment of the oxidation products), chemical, or even biocatalytic approaches. The potential usefulness of the method is not sufficiently demonstrated, as only a limited number of structurally closely related furfuryl alcohol substrates are assayed, while data on the scalability of the method is not provided. In my opinion for this paper to be of interest to the readership of Nature Communications, significant additional work would be needed to unambiguously demonstrate the wide applicability of the reported reaction.

Substrate scope: a larger array of furfuryl alcohols, bearing a variety of different functional groups, needs to be assayed to demonstrate the practical applicability of the method in real-world settings. Ideally, a comparison with existing methods (for example, NBS) would be provided.

Scalability: there is no information provided, as far as I could see, as to the scale that the electrochemical reactions are carried out in. Detailed descriptions of the reaction setup including quantities of reagents and, importantly, of the procedure utilized for the isolation of the products need to be provided in the SI. To demonstrate the synthetic applicability of the reaction, it is important to demonstrate that the reaction can be carried out successfully in a reasonably large scale.

Calculation of yields: In relation to the previous point, isolated yields need to be provided for the various products; HPLC-based yields can be misleading as they do not account for the product purification process. For example, the NMR spectrum of 6-hydroxy-2-methyl-2H-pyran-3(6H)-one appears to contain a significant amount of impurity.

Computational study: as presented, the discussion in the main text only consists of the computational investigation of one possible pathway. As no alternative pathways are discussed, this does not offer any helpful insight to the reader. To better support the mechanism the authors propose, it would be beneficial to move the calculations on alternative pathways from the SI to the main text and expand on this discussion accordingly.

Synthesis of furopyranone: In my opinion, the results presented in this section are not relevant to the rest of the study. The starting material is not made using the electrochemical method the authors report (at least not directly), and the reaction to form furopyranone does not require their special electrocatalyst and instead uses Pt wire as the electrode material. In fact, the same reaction has been carried out simply using NaHCO₃ in water (Org. Lett. 2014, 16, 16, 4284-4287), as the authors themselves point out. The authors criticize the use of stoichiometric NaHCO₃ in the aforementioned study, but their electrochemical method also employs a significant amount (0.1 M) of NaClO₄ as a

supporting electrolyte, which is both more toxic than NaHCO_3 and a strong oxidizer thus presenting a fire hazard.

Reviewer #2 (Remarks to the Author):

This manuscript describes organic synthesis of relevant pharmaceutical products, by using electrochemical procedures, which are attracting enormous interest among chemical and industrial communities. In the introduction the authors begin to refer the capabilities of biomass components that by means of biomass valorization techniques can lead to promising carbon sources that, by properly designed approaches conducts to high-value pharmaceutical and medicinal applications. In this line, the authors refer synthetic strategies that have been established to overpass some issues of biomass as a source of valuable and tested products. By including 37 recent and relevant references in the text they provide excellent information to the potential readers, and simultaneously they contribute to follow their adopted procedures to develop and attain their objectives. The Achmatowicz reaction (AR) is central in the author's proposals, due to its ability to convert furfuryl alcohol into substituted dihydropyranone acetals, which are very useful for many syntheses, apart from the fact that it introduces a chiral center, leading to many asymmetric variants that serve for the development of biologically active products. Several strategies have been used to expand the AR scope, but problems have limited these strategies. Fortunately, the transfer of the chemical synthesis to the electrochemical assisted way, largely reduced some of the reported issues, but even with these electrochemical routes, the authors procedure for attaining their goals, required a novel electrocatalysis approach, by using water as the sole oxygen source. This is reported and discussed in the following sections.

The first step of the work consisted in the photo-induced polymerization of a divinyl bipyridine (dvbpy) which resulted in a web-like ligand film (DVBP) covering the GC anode electrode. This pre-immobilization facilitated the Ni^{2+} metalation, i.e. by this metal cation incorporation it was established a Ni-DVBP electrocatalyst with both homogeneous and heterogeneous advantages. FTIR and Raman spectra, SEM, EDX and XPS confirmed the rough and porous morphology, the Ni-N stretching, the pyridine N, and the composition and valence state of the elements in Ni-DVBP. CV/SWV measurements on the Ni-DVBP/EC electrode in acetonitrile showed a peak at about $-0.3 \text{ V vs Fc}(+/0)$ that the authors assigned to the $\text{Ni}(3+/2+)$ redox couple, occurring at Ni sites immobilized on the electrode. CVs in the presence of furfuryl alcohol also suggested its oxidation. Since the furfuryl polarization can lead to furfural or dihydropyranone, potentiostatic polarizations were carried out in a 3-electrode conventional cell, to calculate the yield, selectivity and faradaic efficiency of dihydropyranone versus furfural, being demonstrated that different working conditions, namely the pH, can identify and quantify the several electrosynthesized products, as further monitored by HPLC. Prolonged electrolysis confirmed the robustness of the NiDVBP electrocatalyst for the AR. This well succeeded process stimulated interest towards other furfuryl alcohol derivatives, being suggested that the electrochemical AR is initiated by the Ni active site olefine unit interaction, as also elucidated by DFT calculations. A following study consisted of checking whether or not it would be possible to electrosynthesize bicyclic units from

hydropyranone. In this context, furapyranone derivative was synthesized by an electrochemical approach. Starting from hydropyranone and acetyl acetone (acac), a Pt electrode was cathodically polarized in an alkaline aqueous electrolyte, being possible to simulate a base-initiated Michael addition followed by cycloacetalization. The conversion of hydropyranone acetate into furapyranone, along 2.5 h of potentiostatic cathodic polarization, determined via HPLC, was observed. The experimental section and the supporting information are also very clear and useful, improving the quality of the paper. In summary, in terms of organic electrosynthesis, this is an innovative and excellent work that that deserves full consideration for the Journal <Nature Communications>. I believe it is a very good contribution by itself, and because it opens a large avenue for the synthesis of other valuable chemicals by organic electrosynthesis.

Point-by-Point Response

Reviewer #1:

The authors report the development of a novel electrocatalyst and its application to the formation of hydropyranones from furfuryl alcohols. This represents, to my knowledge, the first example of a direct electrochemical Achmatowicz reaction, however the same overall transformation is easily achievable using either electrochemical (by post-treatment of the oxidation products), chemical, or even biocatalytic approaches. The potential usefulness of the method is not sufficiently demonstrated, as only a limited number of structurally closely related furfuryl alcohol substrates are assayed, while data on the scalability of the method is not provided. In my opinion for this paper to be of interest to the readership of Nature Communications, significant additional work would be needed to unambiguously demonstrate the wide applicability of the reported reaction.

Substrate scope: a larger array of furfuryl alcohols, bearing a variety of different functional groups, needs to be assayed to demonstrate the practical applicability of the method in real-world settings. Ideally, a comparison with existing methods (for example, NBS) would be provided. Scalability: there is no information provided, as far as I could see, as to the scale that the electrochemical reactions are carried out in. Detailed descriptions of the reaction setup including quantities of reagents and, importantly, of the procedure utilized for the isolation of the products need to be provided in the SI. To demonstrate the synthetic applicability of the reaction, it is important to demonstrate that the reaction can be carried out successfully in a reasonably large scale.

Our response: Following the reviewer's suggestions, we carried out a series of experiments to address the concerns.

Substrate scope: Six furfuryl alcohol derivatives were synthesized to explore the applicability of our Ni-DVBP electrocatalysts for the Achmatowicz reaction. In particular, we focused on the derivatives with various substituents positioned at the hydroxymethyl position and the 5'-position of the furan ring, in that these positions are crucial for the success of the Achmatowicz reaction to take place. As shown in Figure 5 in the revised main text, one methyl, two methyl, or one propyl substituents at the hydroxymethyl group do not prohibit the electrocatalytic Achmatowicz reaction on Ni-DVBP (yield: 87-93%). A methyl substituent at the 5'-position of the furan ring indeed decreased the hydropyranone yield to 82%, probably due to its steric effect in binding to the nickel center and subsequent hydroxide transfer. However, more electron-donating (e.g., methoxy) or more sterically bulky (e.g., para-chlorophenyl) at the 5'-position of the furan ring can essentially prohibit the Achmatowicz reaction. The synthesis of these furfuryl alcohol derivatives and related discussion are included in the revised manuscript.

Comparison with existing methods: In order to compare our electrocatalytic approach with those reported methods, two chemical oxidants, NBS and m-CPBA, were utilized to synthesize hydropyranone and we obtained the hydropyranone yields of 58% and 73%, respectively, which are comparable to literature results (*Angew. Chem. Int. Ed.* **2015**, 54, 8756–8759; *Angew. Chem. Int. Ed.* **2019**, 58, 14715–14723). It should be noted that our electrocatalytic method can achieve

much higher yield (e.g., 96% in Table 3) and no chemical oxidants are required, superior to those existing chemical oxidation methods.

Scalability: Because of the modular nature of electrochemistry, we are able to readily modify our reaction systems to meet various needs of different scales. For instance, herein we introduce three methods to demonstrate that our Ni-DVBP electrocatalyst can be utilized at three different scales with equally excellent performance. Method A is a small-scale experiment and we use HPLC to quantify the yield and selectivity of the final product. Method B is a median-scale experiment with 0.5 mmol (>50 mg) substrate, which is enough for product isolation. Method C employs a flow electrolyzer to prove the suitability of our electrocatalyst for large-scale synthesis. The detailed procedures are described here and also included in the revised supporting information.

Method A: A 20 mL two-compartment cell was equipped with a magnetic stir bar in the anode chamber. Carbon paper loaded with Ni-DVBP and Ag/AgCl were used as the anode and reference electrodes, respectively. A carbon rod was placed in the counter chamber as the counter electrode. To the anode chamber was added 10 mL 0.1 M phosphate buffer (pH 7) and 8.7 μ L furfuryl alcohol. The reaction mixture was then purged with nitrogen gas for 5 minutes. Electrolysis was performed at 1.4 V vs Ag/AgCl. The reaction was monitored by HPLC and the product yield was calculated based on obtained HPLC spectra.

Method B: A 20 mL two-compartment cell was equipped with a magnetic stir bar in the anode chamber. Carbon paper loaded with Ni-DVBP and Ag/AgCl were used as the anode and reference electrodes, respectively. A carbon rod was placed in the counter chamber as the counter electrode. To the anode chamber was added 10 mL 0.1 M phosphate buffer (pH 7) and 0.5 mmol furfuryl alcohol (or its derivatives). For those substrates with low solubility in phosphate buffer, 30% (v/v) THF was added to the electrolyte. The reaction mixture was then purged with nitrogen gas for 5 minutes. Electrolysis was performed at 1.4 V vs Ag/AgCl. Upon full consumption of starting material as determined by thin-layer chromatography (TLC, stained with KMnO_4 solution), the reaction mixture was extracted with CH_2Cl_2 thoroughly. The product solution was concentrated under vacuum and purified with preparative TLC plate (eluted with hexanes/ethyl acetate) to yield the final product.

Method C: A flow electrolyzer was used for large scale synthesis. Three carbon paper electrodes loaded with Ni-DVBP were used as the anode and a Ni foam was used as the counter electrode which were separated by an anion exchange membrane. A solution (250 mL) of 0.1 M phosphate buffer and 50 mM furfuryl alcohol was pumped into the anode chamber and blank phosphate buffer into the counter chamber at a flow rate of 3 mL/h. This was a two-electrode configuration electrolysis and the cell voltage was set at 3.5 V. The reaction mixture from the outlet was collected and extracted with CH_2Cl_2 thoroughly. The product solution was concentrated under vacuum and purified with column chromatography (eluted with hexanes/ethyl acetate) to obtain the final product with a gram-scale yield (1.14 g, yield = 81%).

Calculation of yields: In relation to the previous point, isolated yields need to be provided for the various products; HPLC-based yields can be misleading as they do not account for the product purification process. For example, the NMR spectrum of 6-hydroxy-2-methyl-2H-pyran-3(6H)-one appears to contain a significant amount of impurity.

Our response: We have included the isolated yield of each product from electrolysis and presented their ^1H and ^{13}C NMR spectra in the revised manuscript. After electrolysis, each reaction mixture was extracted with CH_2Cl_2 thoroughly. The product solution was concentrated under vacuum and purified with preparative TLC plate or column chromatography (eluted with hexanes/ethyl acetate) to obtain the purified product.

Computational study: as presented, the discussion in the main text only consists of the computational investigation of one possible pathway. As no alternative pathways are discussed, this does not offer any helpful insight to the reader. To better support the mechanism the authors' propose, it would be beneficial to move the calculations on alternative pathways from the SI to the main text and expand on this discussion accordingly.

Our response: We added detailed discussion of alternative pathways in the revised manuscript (Scheme 2). All the related contents and figures have been highlighted in yellow. It should be noted that the alternative epoxidation pathway requires much higher energetic inputs, which is less likely to take place under our experimental conditions. In addition to the four-coordination structure for the central nickel in Ni-DVBP, six-coordination structure was also considered, which exhibited slightly higher energy demand.

Synthesis of fuopyranone: In my opinion, the results presented in this section are not relevant to the rest of the study. The starting material is not made using the electrochemical method the authors report (at least not directly), and the reaction to form fuopyranone does not require their special electrocatalyst and instead uses Pt wire as the electrode material. In fact, the same reaction has been carried out simply using NaHCO_3 in water (Org. Lett. 2014, 16, 16, 4284-4287), as the authors themselves point out. The authors criticize the use of stoichiometric NaHCO_3 in the aforementioned study, but their electrochemical method also employs a significant amount (0.1 M) of NaClO_4 as a supporting electrolyte, which is both more toxic than NaHCO_3 and a strong oxidizer thus presenting a fire hazard.

Our response: We respectfully disagree with the reviewer's comments. In fact, the synthesis of fuopyranone is a direct example to showcase the synthetic application of hydroxyranone, which is exactly the motivation for us to explore the electrocatalytic Achmatowicz reaction. As included in the Introduction section, the Achmatowicz reaction can be used in the syntheses of many natural products and pharmaceuticals. The synthesis of fuopyranone is a representative example to demonstrate these applications. Therefore, we strongly believe that the synthesis of fuopyranone is relevant to the big picture of this work.

The starting material for the synthesis of fuopyranone was prepared from hydroxyranone (which can be electrocatalytically synthesized following our method) and acetic anhydride with quantitative yield. This esterification step was a group protection step for the following electrochemical synthesis. The reviewer pointed out that the reaction to form fuopyranone did not require special electrocatalyst and instead Pt wire could be used as the electrode material. Indeed, this is the advantage of our electrochemical method for the synthesis of fuopyranone. Organic electrosynthesis does not always need a catalyst. Working electrode with appropriate

potential bias can act as an oxidant (or reductant) with tunable oxidizing (or reducing) power. This is the beauty of organic electrosynthesis.

For this reaction reported in *Org. Lett.* **2014**, 16, 16, 4284-4287, the initiation is the deprotonation of acetylacetone in the presence of inorganic base to produce acetylacetone anion, which only involves proton transfer from acetylacetone to a base. In our system, the working electrode with appropriate negative potential is able to directly reduce acetylacetone to an anion. Therefore, only electricity is consumed in our system and no inorganic base is consumed (or needed).

In our system, NaClO₄ is used as the supporting electrolyte, which will NOT be consumed. In contrast, stoichiometric NaHCO₃ will be consumed in the reported method (*Org. Lett.* **2014**, 16, 16, 4284-4287). In summary, our electrosynthesis of furopyranone highlights the application of the electrocatalytic Achmatowicz reaction on our Ni-DVBP electrocatalyst and exhibits apparent advantages compared to reported methods.

Reviewer #2:

This manuscript describes organic synthesis of relevant pharmaceutical products, by using electrochemical procedures, which are attracting enormous interest among chemical and industrial communities. In the introduction the authors begin to refer the capabilities of biomass components that by means of biomass valorization techniques can lead to promising carbon sources that, by properly designed approaches conducts to high-value pharmaceutical and medicinal applications.

In this line, the authors refer synthetic strategies that have been established to overpass some issues of biomass as a source of valuable and tested products. By including 37 recent and relevant references in the text they provide excellent information to the potential readers, and simultaneously they contribute to follow their adopted procedures to develop and attain their objectives. The Achmatowicz reaction (AR) is central in the author's proposals, due to its ability to convert furfuryl alcohol into substituted dihydropyranone acetals, which are very useful for many syntheses, apart from the fact that it introduces a chiral center, leading to many asymmetric variants that serve for the development of biologically active products.

Several strategies have been used to expand the AR scope, but problems have limited these strategies. Fortunately, the transfer of the chemical synthesis to the electrochemical assisted way, largely reduced some of the reported issues, but even with these electrochemical routes, the authors procedure for attaining their goals, required a novel electrocatalysis approach, by using water as the sole oxygen source. This is reported and discussed in the following sections.

The first step of the work consisted in the photo-induced polymerization of a divinyl bipyridine (dvbpy) which resulted in a web-like ligand film (DVBP) covering the GC anode electrode. This pre-immobilization facilitated the Ni²⁺ metalation, i.e. by this metal cation incorporation it was established a Ni-DVBP electrocatalyst with both homogeneous and heterogeneous advantages. FTIR and Raman spectra, SEM, EDX and XPS confirmed the rough and porous morphology, the Ni-N stretching, the pyridine N, and the composition and valence state of the elements in Ni-DVBP. CV/SWV measurements on the Ni-DVBP/EC electrode in acetonitrile showed a peak at about -0.3 V vs Fc(+0) that the authors assigned to the Ni(3+/2+) redox couple, occurring at Ni sites immobilized on the electrode. CVs in the presence of furfuryl alcohol also suggested its oxidization. Since the furfuryl polarization can lead to furfural or hydroxyfuranone, potentiostatic polarizations were carried out in a 3-electrode conventional cell, to calculate the yield, selectivity and faradaic efficiency of hydroxyfuranone versus furfural, being demonstrated that different working conditions, namely the pH, can identify and quantify the several electrosynthesized products, as further monitored by HPLC. Prolonged electrolysis confirmed the robustness of the Ni-DVBP electrocatalyst for the AR.

This well succeeded process stimulated interest towards other furfuryl alcohol derivatives, being suggested that the electrochemical AR is initiated by the Ni active site olefin unit interaction, as also elucidated by DFT calculations. A following study consisted of checking whether or not it would be possible to electrosynthesize bicyclic units from hydroxyfuranone. In this context, furofuranone derivative was synthesized by an electrochemical approach. Starting from hydroxyfuranone and acetyl acetone (acac), a Pt electrode was cathodically polarized in an alkaline aqueous electrolyte, being possible to simulate a base-initiated Michael addition followed by

cycloacetalization. The conversion of hydroxyranone acetate into fuopyranone, along 2.5 h of potentiostatic cathodic polarization, determined via HPLC, was observed.

The experimental section and the supporting information are also very clear and useful, improving the quality of the paper. In summary, in terms of organic electrosynthesis, this is an innovative and excellent work that that deserves full consideration for the Journal <Nature Communications>. I believe it is a very good contribution by itself, and because it opens a large avenue for the synthesis of other valuable chemicals by organic electrosynthesis.

Our response: We sincerely appreciate that the reviewer supports the acceptance of our work, especially for highlighting the significance and novelty of this study.

REVIEWER COMMENTS

Reviewer #1 (Remarks to the Author):

Overall, the revised substrate screen reveals a limited applicability of the method to substituted furfuryl alcohols. Together with unsatisfactory scholarly presentation (especially in the mechanism section and in the furopyranone synthesis section, see below), I believe this manuscript is not suited to a high-impact broad-readership journal such Nature Communications and would be more suited to a more specialized journal. Detailed comments follow:

Substrate screen: Only a small number of additional substrates were assayed, and it appears that more complex substrates are not well tolerated by the method. A comparison with established chemical oxidants is only performed for the simplest case (hydroxyacetone itself), and therefore it does not provide any useful comparative information.

An additional note: compound 2f in Figure 5 is not the expected product, as it would readily collapse to the lactone.

Mechanism : Intermediate 6 appears relatively stable and is not typically invoked as an intermediate in Achmatowicz reactions; also, ref 50 does not support the claim that it will “readily rearrange to hydroxyacetone”, as the authors claim. Furthermore, the arrow-pushing in Scheme 2 b is wrong (it would lead to a hypervalent oxygen and not to the aldehyde structure shown)

Furopyranone synthesis: My comments on the last section remain the same; I believe it is not novel (especially given the precedent in Org. Lett. 2014, 16, 16, 4284-4287) and additionally not relevant to the rest of the study. The claim by the authors that the supporting electrolyte is not consumed, unlike the base in the aforementioned paper, is not of practical importance, as one does not typically recover the supporting electrolyte from an electrolysis experiment.

Point-by-Point Response

Reviewer #1:

Overall, the revised substrate screen reveals a limited applicability of the method to substituted furfuryl alcohols. Together with unsatisfactory scholarly presentation (especially in the mechanism section and in the furopyranone synthesis section, see below), I believe this manuscript is not suited to a high-impact broad-readership journal such Nature Communications and would be more suited to a more specialized journal. Detailed comments follow:

Substrate screen: Only a small number of additional substrates were assayed, and it appears that more complex substrates are not well tolerated by the method. A comparison with established chemical oxidants is only performed for the simplest case (hydroxyfuranone itself), and therefore it does not provide any useful comparative information.

An additional note: compound **2f** in Figure 5 is not the expected product, as it would readily collapse to the lactone.

Our response: Following the reviewer's comment, we extended the substrate screening to include to more furfural alcohol derivatives synthesized by our own group, as they are not commercially available. In particular, we also applied our electrocatalytic strategy to those furfural alcohol substrates which could be readily utilized for the synthesis of pharmaceutically relevant natural products.

As shown in **Figure R1 (Fig. 5)** in the revised manuscript, 13 substrates were studied to demonstrate the applicability of our electrocatalytic Achmatowicz reaction method. Substituents of different steric hindrance at the hydroxymethyl position (**1a-1f**) were assessed. High yields of their corresponding hydroxyfuranone products were obtained, ranging from 82% to 93%, which indicates the consistency and applicability of our method. In addition, substrates with varying substituents of different electronic effect were also investigated. For instance, a *p*-fluorophenyl substituent (**1i**) exhibited stronger electron-withdrawing effect relative to phenyl (**1g**) and *p*-tolyl (**1h**) groups, resulting in a slightly lower yield of the corresponding hydroxyfuranone product, 76% (**2i**) vs 91-89% (**2g-2h**).

It should be noted that a methyl substituent at the 5'-position of the furan was also considered and a decent yield of 82% was obtained for the corresponding hydroxyfuranone product **2j**. However, a much bulky substituent *p*-chlorophenyl at the same position prohibited the formation of the desirable hydroxyfuranone **2k** using our electrocatalytic approach, strongly highlighting the importance of accessing the 5'-position of furan for the success of our electrocatalytic Achmatowicz reaction. If *m*-CPBA was employed as a chemical oxidant, **2k** was produced with a yield of 75%, which suggests that our electrocatalytic mechanism is strikingly distinctive from conventional mechanisms utilizing chemical oxidants.

To further demonstrate the applicability of our electrocatalytic Achmatowicz strategy towards highly valuable reactions, we evaluated furfural alcohol substrates with unique functional groups that are essential in the synthesis of natural products. For example, the Achmatowicz reaction has been adopted to synthesize aspergillide and its analogues. The synthesis of **2l** is critical to introduce the six-member ring and the ester group plays a vital role to the ring-closure step in the synthesis of the final product (-)-Aspergillide A. Our electrocatalytic Achmatowicz strategy was successfully used in the synthesis of **2l** and the yield was comparable

with that obtained using *m*-CPBA as the chemical oxidant. In addition, an oxidation sensitive vinyl group was well tolerated in our electrocatalytic Achmatowicz reaction method in the preparation of **2m**, paving the way for the synthesis of halichondrins. These two successful examples further showcase the promise of our electrocatalytic Achmatowicz reaction as a green and facile approach for practical and useful organic transformations.

Following the reviewer's comments, we also established the comparison of our electrocatalytic approach with conventional chemical oxidation approach using *m*-CPBA as an oxidant. The yields of the hydropyranones obtained from the utilization of *m*-CPBA are also included in **Figure R1**. It is apparent that our electrocatalytic Achmatowicz strategy was able to produce better (or comparable) yields in most cases, except the case of **1k**, due to the unique mechanism of our electrocatalytic approach.

Figure R1. The Achmatowicz reaction yields of furfuryl alcohol derivatives obtained on Ni-DVBP at 1.4 V vs Ag/AgCl in 0.1 M phosphate buffer of pH 7. The corresponding product yields obtained through *m*-CPBA oxidation are also shown here.

Mechanism: Intermediate **6** appears relatively stable and is not typically invoked as an intermediate in Achmatowicz reactions; also, ref 50 does not support the claim that it will “readily rearrange to hydropyranone”, as the authors claim. Furthermore, the arrow-pushing in Scheme 2 b is wrong (it would lead to a hypervalent oxygen and not to the aldehyde structure shown).

Our response: Despite of great efforts of our group in trying to synthesize intermediate **6**, we were not able to synthesize it successfully. Based on literature reading, we obtained the information of its close analogue: 2-furanol, which does not have the hydroxymethyl group in **6**. The molecular orbital study indicated that 2-furanol is not a preferred structure^[1] and even may not exist in practice^[2-3]. Therefore, we speculated that intermediate **6** is not stable, especially under our electrocatalytic condition. We sincerely appreciate the reviewer’s concerns about our original hypothesis of its rearrangement steps. Herein, we redrew the rearrangement route in **Figure R2** (Scheme 2b in the revised manuscript). Once the intermediate **6** is generated, hydroxide-mediated ring opening could take place under our electrocatalytic condition. A hydroxide anion will attack the 2'-position of **6** and hence it will rearrange to intermediate **8**, which is similar to the highlighted intermediate in the widely-accepted Br/MeOH-driven Achmatowicz reaction shown in **Figure R3**^[4,5]. The ring-opening intermediate **9** will be formed after the release of the hydroxide anion. The final ring-closure step will lead to the production of the final hydropyranone.

Figure R2. Proposed rearrangement scheme of intermediate **6** to hydropyranone in our electrocatalytic Achmatowicz reaction.

Figure R3. Proposed mechanism of the Br/MeOH mediated Achmatowicz reaction.

[1] Bodor, N.; Dewar, M. J. S.; Harget, A. J. *J. Am. Chem. Soc.* **1970**, 92, 2929.

[2] Näsman, J. H.; Pensar, K. G. *Synthesis* **1985**, 786.

[3] Boukouvalas, J. α,β -Butenolide. DOI:10.1002/047084289X.rb346.

[4] Achmatowicz, O.; Bukowski, P.; Szechner, B.; Zwierzchowska, Z.; Zamojski, A. *Tetrahedron* **1971**, 27, 1973.

[5] <https://en.chem-station.com/reactions-2/2015/03/achmatowicz-reaction.html>.

Furopyranone synthesis: My comments on the last section remain the same; I believe it is not novel (especially given the precedent in *Org. Lett.* 2014, 16, 16, 4284-4287) and additionally not relevant to the rest of the study. The claim by the authors that the supporting electrolyte is not consumed, unlike the base in the aforementioned paper, is not of practical importance, as one does not typically recover the supporting electrolyte from an electrolysis experiment.

Our response: We respectfully disagree with the reviewer's comments. As the title of our manuscript suggests, the aim of this project is to electrocatalytically synthesize heterocycles from furfuryl alcohols. In order to demonstrate this idea, it should not be restricted to the synthesis of hydroxyranones. Instead, it is equally appealing to obtain complex heterocycles starting from furfural alcohol substrates. In fact, the synthesis of furopyranone is a great manifest of the synthetic application of hydroxyranone, which is obtained from our electrocatalytic Achmatowicz reaction. Overall, we aim to report green electrocatalytic methods for the synthesis of two valuable target products, with coherent logic between their synthetic routes.

Furthermore, the advantages of our furopyranone electrosynthesis compared to the report in *Org. Lett.* 2014, 16, 16, 4284-4287 are apparent. First, the only chemical consumed besides reactants in the electrosynthesis of furopyranone is water, which is considerably abundant and inexpensive relative to NaHCO_3 used in the reported method. Secondly, O_2 is generated on the anode in our electrolysis which can be collected as a useful chemical, while for the *Org. Lett.* report Na_2CO_3 was produced in solution.

We hold different opinion regarding the reviewer's comment on electrolyte recovery. Electrolyte recovery is not only an important process in laboratory, but also in large-scale applications. For example, Würde et al reported a procedure of tetra(n-butyl)ammonium hexafluorophosphate recovery from used electrolyte solutions (*Current Separations*, 1996, 15, 53-56). It should be noted that electrolyte recovery in our system is quite straight forward because our electrolytes are aqueous solutions while our products could be readily extracted with organic solvents, leaving the electrolyte solutions intact. Extraction is common practice in organic synthesis which is also utilized in the reported method (*Org. Lett.* 2014, 16, 16, 4284-4287). Thus, the utilization of supporting electrolyte is not a disadvantage at all in our system.

REVIEWER COMMENTS

Reviewer #1 (Remarks to the Author):

Overall:

The expanded substrate scope of the study and the comparison with mCPBA does put the method in a better context from a synthetic applicability point of view, indicating that for specific substrates (i.e. ones not bearing a bulky substituent at the 5 position), the method is comparable to established methods (mCPBA oxidation). However, my other comments remain unaddressed (see specific explanations below). I have serious concerns about the validity of the mechanism proposed, and as a consequence also about the proposed direct interaction of Ni with the substrate, which is one of the main conclusions of the work. Also and as I noted previously, the last section of the manuscript is not relevant to the rest of the work, and including it greatly impacts the scholarly presentation of the work negatively.

As noted previously, the Achmatowicz reaction is easily achievable via various means, and the main advantage of the current approach lies in its “greener” nature; while a valid advantage, in my view it is more relevant to a specialized audience. Taking all of these aspects into consideration, I unfortunately have to maintain my position that this manuscript is not suitable for publication in Nature Communications and would be better suited to a more specialized electrochemistry or green chemistry journal (ideally after revision of the mechanistic aspect).

Substrate scope:

It needs to be noted that for the mCPBA reaction of substrate 1a, the yield reported by the authors (58%) is significantly lower than literature mCPBA oxidations of the same substrate (90% in J. Org. Chem., 1997, 62, 5, 1257 – 1263 and 88% in Carbohydrate Research, 1991, 222, 163 – 172 and in Tetrahedron, 1993, 49, 40, 8999 – 9018 among others).

Mechanism:

My comments on the mechanism depicted by the authors remain the same, as they have not been addressed by their redrawing of the mechanism: the arrow-pushing from compound 7 to compound 8 is wrong and would not lead to the structure depicted as 8; formally it would lead to a hypervalent oxygen anion, which is of course not a viable species. Furthermore, when the correct arrow pushing is written out (arrows pushed in the opposite direction, starting with deprotonation of the O-H bond), the 2 position of the furan ring is at the carboxylic acid/ester oxidation level while the 5 position is at the alcohol oxidation level; in an Achmatowicz reaction both positions end up at the aldehyde/ketone oxidation level, as exemplified clearly by intermediate 9 (which is of course a known intermediate in Achmatowicz reactions).

It should be noted that even the attack of hydroxide on intermediate 7 is very dubious, as it results in the dearomatization of a furan – instead in the example of a bromine-mediated Achmatowicz reaction shown in R3 in the authors rebuttal, attack of the hydroxide anion is driven by the opening of the bromonium ion (and the substrate of the attack is of course not aromatic).

In my mind therefore the intermediacy of a putative compound 6 in an Achmatowicz reaction is not

clearly evident from the evidence presented, which also puts the rest of the mechanistic study into question as it relies on the generation of compound 6. A mechanism closer to Mechanism 2 in the manuscript (which the authors disfavor based on their DFT studies) seems to me much more likely from a mechanistic point of view. Whether this happens exactly as shown in Mechanism 2 (through the intermediacy of peroxo nickel compound 3') or through some other process, I cannot say with the data presented.

Related to that last note, this also raises the question of whether the reaction is mediated by electrochemically generated peroxide, and not through a direct interaction with Ni as the authors propose. In connection to this, the authors do see some product formation when using DVBP (without Ni) and carbon paper. They ascribe this to a different mechanism operating, but present no evidence for that (except perhaps for the observation that the reaction does not work with furans substituted in the 5- position, which the authors attribute to steric hindrance – however these specific substrates were not tested with DVBP by itself and with carbon paper in order to directly compare. Even if they were, this is unlikely to be sufficient proof by itself for the steric hindrance argument).

Furopyranose synthesis:

My comments remain the same. I repeat here, from my review of the first version of the manuscript: "In my opinion, the results presented in this section are not relevant to the rest of the study. The starting material is not made using the electrochemical method the authors report (at least not directly), and the reaction to form furopyranone does not require their special electrocatalyst and instead uses Pt wire as the electrode material. In fact, the same reaction has been carried out simply using NaHCO₃ in water (Org. Lett. 2014, 16, 16, 4284-4287), as the authors themselves point out. The authors criticize the use of stoichiometric NaHCO₃ in the aforementioned study, but their electrochemical method also employs a significant amount (0.1 M) of NaClO₄ as a supporting electrolyte, which is both more toxic than NaHCO₃ and a strong oxidizer thus presenting a fire hazard."

If the point of this section is to demonstrate the usefulness of hydroxypyronone structures towards further, operationally simple green reactions, it would have been sufficient to cite the Org. Lett. 2014, 16, 16, 4284-4287 example. The contents of this section belong to a separate paper, and will then need to be evaluated against the backdrop of the aforementioned Org. Lett. precedent. To answer the authors' points in their rebuttal:

"Würde et al reported a procedure of tetra(n-butyl)ammonium hexafluorophosphate recovery from used electrolyte solutions (Current Separations, 1996, 15, 53-56)":

This is only one example and concerns a different supporting electrolyte than the one used in the paper. "It should be noted that electrolyte recovery in our system is quite straight forward... Extraction is common practice in organic synthesis":

To use this argument, the authors would need to back this statement up by performing the described experiment (recovering the aqueous solution after extraction of the product, repeating the electrolysis using the recovered solution/electrolyte, always getting consistent yields after repeated cycles of this process). In my view it is not a worthwhile effort to do this for this manuscript, as even if successful it would still be only a minor advantage over the Org. Lett. 2014 method, and most importantly, it would still lack a connection with the main study of the paper. As stated, I believe this section belongs to separate paper to be evaluated independently.

Point-by-Point Response

Reviewer #1:

Overall:

The expanded substrate scope of the study and the comparison with mCPBA does put the method in a better context from a synthetic applicability point of view, indicating that for specific substrates (i.e. ones not bearing a bulky substituent at the 5 position), the method is comparable to established methods (mCPBA oxidation). However, my other comments remain unaddressed (see specific explanations below). I have serious concerns about the validity of the mechanism proposed, and as a consequence also about the proposed direct interaction of Ni with the substrate, which is one of the main conclusions of the work. Also and as I noted previously, the last section of the manuscript is not relevant to the rest of the work, and including it greatly impacts the scholarly presentation of the work negatively.

As noted previously, the Achmatowicz reaction is easily achievable via various means, and the main advantage of the current approach lies in its “greener” nature; while a valid advantage, in my view it is more relevant to a specialized audience. Taking all of these aspects into consideration, I unfortunately have to maintain my position that this manuscript is not suitable for publication in Nature Communications and would be better suited to a more specialized electrochemistry or green chemistry journal (ideally after revision of the mechanistic aspect).

Our response: We appreciate the reviewer’s comments very much, but we have to respectfully disagree with his/her opinion that our work is more relevant to a specialized audience. In fact, upgrading of biomass-derived feedstocks to highly valuable products with potential applications in the pharmaceutical fields is a very appealing yet much underexplored direction. Even though the Achmatowicz reaction has been reported using various chemical oxidants, these chemical oxidation approaches are neither green nor sustainable. We noted that several electrochemical approaches have also been reported. However, as discussed in the introduction section of the revised manuscript, they usually require very high voltage, bromine source, alcohols (e.g., methanol and ethanol), and hydrolysis post-treatment. It would be very desirable to develop an **electrocatalytic** approach operating under **ambient condition** and preferably using **water** as the oxygen source as well as solvent.

In this work, we report an electrocatalyst with molecular nickel active sites immobilized in a polymeric ligand film, which exhibits superior selectivity, yield, Faradaic efficiency, and robustness in driving the Achmatowicz reaction electrocatalytically. We believe our electrocatalytic system possesses the following advantages: (1) our heterogenized Ni-DVBP electrocatalyst is able to function under environmentally benign condition; (2) water is used not only as the reaction solvent but also the oxygen source. Toxic and/or expensive chemical oxidants are completely circumvented, hence no other byproducts or wastes are generated, resulting in easy separation and purification of the product and high atom economy; (3) H₂ is produced at the counter electrode simultaneously, which is another valuable product; and (4) our photo-induced ligand polymerization provides a convenient method to prepare many other 1st-row transition metal-based electrocatalytic systems with unsaturated coordination spheres, which are hard to achieve under conventional homogeneous conditions, therefore numerous new applications could be envisaged.

Given the rapidly increasing interest in renewable energy catalysis, biomass valorization, and organic electrochemistry, we are confident that our work will attract the broad audience of *Nature Communications*.

Substrate scope:

It needs to be noted that for the mCPBA reaction of substrate 1a, the yield reported by the authors (58%) is significantly lower than literature mCPBA oxidations of the same substrate (90% in *J. Org. Chem.*, 1997, 62, 5, 1257 – 1263 and 88% in *Carbohydrate Research*, 1991, 222, 163 – 172 and in *Tetrahedron*, 1993, 49, 40, 8999 – 9018 among others).

Our response: We performed the chemical oxidation reaction of substrate **1a** using *m*-CPBA at a larger scale (0.9 g furfuryl alcohol) and the yield of hydropyranone we could obtain was 82%. This information has been added to the revised manuscript.

Mechanism:

My comments on the mechanism depicted by the authors remain the same, as they have not been addressed by their redrawing of the mechanism: the arrow-pushing from compound 7 to compound 8 is wrong and would not lead to the structure depicted as 8; formally it would lead to a hypervalent oxygen anion, which is of course not a viable species. Furthermore, when the correct arrow pushing is written out (arrows pushed in the opposite direction, starting with deprotonation of the O-H bond), the 2 position of the furan ring is at the carboxylic acid/ester oxidation level while the 5 position is at the alcohol oxidation level; in an Achmatowicz reaction both positions end up at the aldehyde/ketone oxidation level, as exemplified clearly by intermediate 9 (which is of course a known intermediate in Achmatowicz reactions).

It should be noted that even the attack of hydroxide on intermediate 7 is very dubious, as it results in the dearomatization of a furan – instead in the example of a bromine-mediated Achmatowicz reaction shown in R3 in the authors rebuttal, attack of the hydroxide anion is driven by the opening of the bromonium ion (and the substrate of the attack is of course not aromatic).

In my mind therefore the intermediacy of a putative compound 6 in an Achmatowicz reaction is not clearly evident from the evidence presented, which also puts the rest of the mechanistic study into question as it relies on the generation of compound 6. A mechanism closer to Mechanism 2 in the manuscript (which the authors disfavor based on their DFT studies) seems to me much more likely from a mechanistic point of view. Whether this happens exactly as shown in Mechanism 2 (through the intermediacy of peroxo nickel compound 3') or through some other process, I cannot say with the data presented.

Our response: We are extremely grateful of the reviewer's insightful comments and we are sorry for mistaken steps proposed in the original Mechanism 1. After carefully considering all the possible reaction steps and recalculation, we propose a new Mechanism 1 shown in Scheme 2. The following discussion related to this new mechanism has been added to the revised manuscript.

“To aid the understanding of the electrocatalytic Achmatowicz reaction on Ni-DVBP, density functional theory (DFT) calculations at the B3LYP/6-31G(d) level were performed. Based on the FT-IR and XPS data, we proposed a molecular model of a mononuclear Ni coordinated with one 2,2'-bipyridyl and two aqua ligands, [(bpy)Ni^{II}(H₂O)₂]²⁺, to mimic the resting state of the Ni

active sites in Ni-DVBP. With $[(bpy)Ni^{II}(H_2O)_2]^{2+}$ as the starting model, our DFT calculation results indicate that it is thermodynamically favorable for furfuryl alcohol to replace one of the aqua ligands with an energy decrease of -0.21 eV (intermediate **2** shown in Scheme 2)."

"DFT calculation was also performed to probe the subsequent oxidation and deprotonation of intermediate **2**, which triggered the interaction between the Ni^{II} -OH center and the 5'-position carbon in the bound furfuryl alcohol as shown as intermediate **3** in Scheme 2. This step requires an energy input of 0.79 eV. Followed by the coordination of another solvent water molecule to Ni in **4**, the hydroxide group originally ligated on Ni migrates to the 5' position of furfuryl alcohol whose alcohol group is liberated from the Ni center. This step is energetically favorable since the energy of **4** is 0.37 eV lower than that of **3**. The second oxidation and deprotonation occur to **4** would further drive the rearrangement process (Supplementary Fig. 6) to liberate intermediate **5** and back to the original state of $[(bpy)Ni(H_2O)_2]^{2+}$ after accepting a solvent water molecule. This step is very energetically downhill from 0.21 eV to -0.60 eV. It should be noted that intermediate **5** is a well anticipated intermediate in the Achmatowicz reaction, which will undergo the final ring closure step to furnish the formation of hydroxypropanone."

Scheme 2 Mechanism 1: (a) DFT calculated energy profile of catalytic intermediates using $[(bpy)Ni(H_2O)_2]^{2+}$ as a catalyst model. The optimized chemical structure of each intermediate is

also included. Mechanism 2: The chemical structure of each intermediate using the same catalyst model but following the oxygen atom transfer pathway.

The electron pushing process in intermediate **4** to release the original catalyst model **1** and intermediate **5** is shown in the Supplementary Fig. 6.

Supplementary Fig. 6 (a) Proposed electron pushing process for the generation of intermediate **5** from **4**. (b) DFT-calculated energy profile of Mechanism 1.

Related to that last note, this also raises the question of whether the reaction is mediated by electrochemically generated peroxide, and not through a direct interaction with Ni as the authors propose. In connection to this, the authors do see some product formation when using DVBP (without Ni) and carbon paper. They ascribe this to a different mechanism operating but present no evidence for that (except perhaps for the observation that the reaction does not work with furans substituted in the 5- position, which the authors attribute to steric hindrance – however these specific substrates were not tested with DVBP by itself and with carbon paper in order to directly compare. Even if they were, this is unlikely to be sufficient proof by itself for the steric hindrance argument).

Our response: In order to probe the interaction between Ni-DVBP and furfuryl alcohol, we collected and compared the IR spectra of free furfuryl alcohol, Ni-DVBP, and Ni-DVBP exposed to furfuryl alcohol. The following discussion has been added to the revised manuscript.

“The interaction between furfuryl alcohol and Ni-DVBP was investigated by comparing the corresponding IR spectra of furfuryl alcohol (Fig. 6a), Ni-DVBP (Supplementary Fig. 7), and Ni-DVBP treated with furfuryl alcohol (Ni-DVBP+furfuryl alcohol, Fig. 6b). Specifically, Ni-DVBP was immersed in an aqueous solution of furfuryl alcohol overnight and subsequently rinsed thoroughly with water. Next, it was dipped in water under stirring for 5 h to remove any loosely adsorbed furfuryl alcohol and finally Ni-DVBP+furfuryl alcohol was obtained for IR measurement. As shown in Fig. 6a, the major IR features of free furfuryl alcohol at 811.4, 909.9, 1002.5, and 1189.2 cm^{-1} could be ascribed to its HC2C3H rocking, C1O stretching, C5O bending, and C1C5 stretching, according to the literature assignments⁵¹. In the IR spectrum of Ni-DVBP+furfuryl alcohol (Fig. 6b), these furfuryl alcohol features also appeared but were shifted to 794.4, 922.2, 1012.6, and 1168 cm^{-1} , respectively. In addition, the pyridine C=N bending peak of Ni-DVBP was also shifted from 1464.0 cm^{-1} (Supplementary Fig. 7) in Ni-DVBP to 1475.4 cm^{-1} (Fig. 6b) in Ni-DVBP+furfuryl alcohol⁵². Overall, these IR spectra demonstrate that furfuryl alcohol is indeed bound within Ni-DVBP and the mutual influence of their IR features strongly support the ligation of furfuryl alcohol to the Ni center in Ni-DVBP, in agreement with the above DFT calculated formation of intermediate 2.”

Fig. 6 IR spectra of (a) furfuryl alcohol with the assignments of major features together with the numbered chemical structure of furfuryl alcohol and (b) NiDVBP+furfuryl alcohol with the marked shifts of furfuryl alcohol features. ν , bond stretching; δ , bending; γ , rocking; τ , torsion. We further conducted a series of electrolysis experiments on Ni-DVBP and blank carbon electrode at different potentials to elucidate the oxidation mechanism under different conditions. The following discussion has been added to the revised manuscript.

“Given the calculated difference in energy requirement between Mechanisms 1 and 2, we reasoned that the preferred oxidation pathway of the electrocatalytic Achmatowicz reaction would be highly dependent on applied potential. Therefore, a series of electrocatalytic oxidation of furfuryl alcohol on Ni-DVBP and blank carbon paper were conducted under the same condition at 1.1, 1.2, 1.3, and 1.4 V vs Ag/AgCl (Supplementary Fig. 9). Fig. 7 and Supplementary Table 1 present the corresponding yields of hydroxyphenone obtained on Ni-DVBP and carbon paper. As indicated in Fig. 3d, 1.1 V vs Ag/AgCl is the onset potential of furfuryl alcohol oxidation on Ni-DVBP. Nearly 100% conversion of furfuryl alcohol and decent hydroxyphenone yields of 79% and 87% were obtained on Ni-DVBP at 1.1 and 1.2 V, respectively. In sharp contrast, electrolysis conducted on blank carbon paper resulted in negligible furfuryl alcohol conversion and hydroxyphenone yield at these applied potentials. Along the increase of applied potential to 1.3 and 1.4 V, the resulting hydroxyphenone yields reached 92% and 96%, respectively, on Ni-DVBP. Because carbon paper started to exhibit water oxidation capability beyond 1.3 V vs Ag/AgCl (Fig. 3c), it is reasonable to anticipate that furfuryl alcohol oxidation could also take place on carbon paper at 1.3 and 1.4 V. Indeed, a small amount of hydroxyphenone (<20%) was detected when carbon paper was employed as the working electrode. Further increasing the applied potential to 1.8 V vs Ag/AgCl, wherein vigorous O_2 evolution could be observed on both Ni-DVBP and carbon paper, the addition of furfuryl alcohol led to the formation of hydroxyphenone with 93% and 65% yields on Ni-DVBP and carbon paper, respectively, albeit accompanied by much lower Faradaic efficiencies (67% on Ni-DVBP and 53% on carbon paper). At such a high applied potential, the calculated Mechanism 2 could be operative, even though the same active species (e.g., Ni-oxyhydroxide) is also very active for O_2 evolution.”

Fig. 7 Hydroxyphenone (HPO) yields in 0.1 M phosphate buffer utilizing Ni-DVBP and carbon paper as working electrodes at different potentials.

With regard to substrates with bulky substituents at the 5'-position of the furan ring, we performed the electrolysis of **1k** on Ni-DVBP at 1.8 V vs Ag/AgCl and the following discussion has been added to the revised manuscript.

*“Encouraged by the above potential-dependent electrolysis results of Ni-DVBP, we went back to reinvestigate the oxidation of substrate **1k** whose p-chlorophenyl substituent at the 5'-position prohibited its oxidation on Ni-DVBP at 1.4 V vs Ag/AgCl. When the applied potential was 1.8 V vs Ag/AgCl, a high yield (84%) of the corresponding hydroxyranone product **2k** was obtained. Such a drastic difference in the oxidation of substrate **1k** at 1.4 and 1.8 V indicate that two distinct oxidation mechanisms are involved. At low applied potentials (1.1 – 1.4 V Ag/AgCl), intramolecular hydroxide transfer from the nickel center to the 5'-position of furfuryl alcohol is the dominant pathway, which is strongly influenced by the bulky substituents at the 5'-position. However, at high applied potential (e.g., 1.8 V vs Ag/AgCl), direct oxygen atom transfer likely takes place between Ni-DVBP (and electrode) and furfuryl alcohol. Nevertheless, much lower Faradaic efficiency is obtained because of the severe competition of the O₂ evolution process.”*

Furopyranone synthesis:

My comments remain the same. I repeat here, from my review of the first version of the manuscript: “In my opinion, the results presented in this section are not relevant to the rest of the study. The starting material is not made using the electrochemical method the authors report (at least not directly), and the reaction to form furopyranone does not require their special electrocatalyst and instead uses Pt wire as the electrode material. In fact, the same reaction has been carried out simply using NaHCO₃ in water (Org. Lett. 2014, 16, 16, 4284-4287), as the authors themselves point out. The authors criticize the use of stoichiometric NaHCO₃ in the aforementioned study, but their electrochemical method also employs a significant amount (0.1 M) of NaClO₄ as a supporting electrolyte, which is both more toxic than NaHCO₃ and a strong oxidizer thus presenting a fire hazard.”

If the point of this section is to demonstrate the usefulness of hydroxypyranone structures towards further, operationally simple green reactions, it would have been sufficient to cite the Org. Lett. 2014, 16, 16, 4284-4287 example. The contents of this section belong to a separate paper, and will then need to be evaluated against the backdrop of the aforementioned Org. Lett. precedent. To answer the authors' points in their rebuttal:

“Würde et al reported a procedure of tetra(n-butyl)ammonium hexafluorophosphate recovery from used electrolyte solutions (Current Separations, 1996, 15, 53-56)”:

This is only one example and concerns a different supporting electrolyte than the one used in the paper.

“It should be noted that electrolyte recovery in our system is quite straight forward... Extraction is common practice in organic synthesis”:

To use this argument, the authors would need to back this statement up by performing the described experiment (recovering the aqueous solution after extraction of the product, repeating the electrolysis using the recovered solution/electrolyte, always getting consistent yields after repeated cycles of this process). In my view it is not a worthwhile effort to do this for this manuscript, as even if successful it would still be only a minor advantage over the Org. Lett. 2014 method, and most importantly, it would still lack a connection with the main study of the paper. As stated, I believe this section belongs to separate paper to be evaluated independently.

Our response: Following the reviewer's suggestion, we removed the content associated with the furopyranone synthesis.

REVIEWERS' COMMENTS

Reviewer #1 (Remarks to the Author):

The additional experiments in the revised manuscript do address my comments on the technical side of the manuscript. The mechanistic discussion has been revised to propose compound 5 as the intermediate towards the final product, which is in fact a viable intermediate in an Achmatowicz reaction. Additional experiments have been carried out that provide support for the proposed direct interaction with Nickel. Also the section on the furopyranone synthesis has been removed, as I suggested.

I have only one additional minor comment on the revised mechanistic discussion in scheme 2, mechanism 2: As written, it is unclear where the second oxygen on the peroxide ligand comes from in structure 3', as no oxygen atom appears on the arrow 2' -> 3'; presumably water is involved but does not appear as a reactant.

Point-by-Point Response

Reviewer #1 (Remarks to the Author):

The additional experiments in the revised manuscript do address my comments on the technical side of the manuscript. The mechanistic discussion has been revised to propose compound 5 as the intermediate towards the final product, which is in fact a viable intermediate in an Achmatowicz reaction. Additional experiments have been carried out that provide support for the proposed direct interaction with Nickel. Also the section on the furopyranone synthesis has been removed, as I suggested. I have only one additional minor comment on the revised mechanistic discussion in scheme 2, mechanism 2: As written, it is unclear where the second oxygen on the peroxide ligand comes from in structure 3', as no oxygen atom appears on the arrow 2' \rightarrow 3'; presumably water is involved but does not appear as a reactant.

Our response: Water is indeed involved in step two to form Ni-oxyhydroxide. A revised scheme is shown below and added to Fig. 7 in the revised manuscript.